# Retrieval of top-of-atmosphere fluxes from combined EarthCARE LiDAR, imager and broadband radiometer observations: the BMA-FLX product

Almudena Velázquez Blázquez[1], Carlos Domenech[2], Edward Baudrez[1], Nicolas Clerbaux[1], Carla Salas Molar[2], and Nils Madenach[3]

[1]Royal Meteorological Institute of Belgium, Brussels, Belgium
[2]GMV, Madrid, Spain
[3]Freie Universität Berlin, Germany

**Correspondence:** Almudena Velázquez-Blázquez (almudena.velazquez@meteo.be)

**Abstract.** The Earth Cloud, Aerosol and Radiation Explorer (EarthCARE) satellite mission is expected to provide new insights into aerosols, clouds, and radiation. The satellite's payload include four instruments designed to synergistically retrieve vertical profiles of clouds and aerosols, along with the atmospheric radiation data. This will enable the determination of atmospheric heating rates and top-of-atmosphere radiances and fluxes. This paper focuses on the BMA-FLX processor, specifically created, developed, and validated to retrieve thermal and solar top-of-atmosphere radiative fluxes from longwave and shortwave radiances, measured along-track by the EarthCARE Broadband Radiometer (BBR) instrument. These radiances are co-registered either at the surface or, in cloudy conditions, at the radiatively most significant vertical layer of the atmosphere (reference level). The Multi-Spectral Imager (MSI) and Atmospheric LiDAR (ATLID) on-board EarthCARE support cloud identification, while meteorological data from the European Centre for Medium-Range Weather Forecasts provide the surface and atmospheric necessary information. In the BMA-FLX processor, flux is estimated independently for each BBR view using different approaches for the longwave and shortwave radiances. A combined flux, derived from co-registered radiances at the reference level, is provided as the best estimate for a given scene. The radiance-to-flux conversion algorithms have been successfully validated through end-to-end verification using L1 and L2 synthetic data for three EarthCARE orbits. In general, a good agreement is found between the retrieved fluxes and the model truth, with RMSEs varying between $7 \ \mathrm{Wm^{-2}}$ and $18 \ \mathrm{Wm^{-2}}$ for the solar fluxes and lower than $6 \ \mathrm{Wm^{-2}}$ for the thermal fluxes. The BMA-FLX's objective is to achieve radiative closure for EarthCARE with solar and thermal fluxes within $10 \ \mathrm{Wm^{-2}}$.

## 1 Introduction

The accurate representation of the complex interplay between aerosols, clouds, and radiation in climate models remains a significant uncertainty in climate projections (Boucher et al., 2013). The Earth Cloud Aerosol and Radiation Explorer (Earth-CARE) mission, a collaboration between the European Space Agency (ESA) and the Japan Aerospace Exploration Agency (JAXA), aims to enhance our understanding of the interaction between clouds and aerosols and their impact on Earth's radia-

tion balance (Illingworth et al., 2015; Wehr et al., 2023; Eisinger et al., 2024). EarthCARE is the first platform to measure both vertical structure and horizontal distribution of cloud and aerosol fields simultaneously from one platform using one passive and two active instruments. Additionally, the emitted longwave and reflected shortwave radiation are measured. The four sci-
entific instruments observe the Earth from a sun-synchronous polar orbit crossing the equator in the early afternoon to optimize observation of convective cloud systems in the tropics. The two active instruments, cloud profile radar (CPR) and atmospheric LiDAR (ATLID), provide information on the vertical structure of the micro- and macro-physical properties of clouds, aerosols, and hydro meteors (Kollias et al., 2022; Irbah et al., 2023; Mroz et al., 2023; Donovan et al., 2023; van Zadelhoff et al., 2023b). The passive instrument, the multi-spectral imager (MSI), gives horizontal context to the observed scene (Wandinger et al.,
2023; Docter et al., 2023; Hünerbein et al., 2023, 2024; Haarig et al., 2023) and is used to create 3D scenes (Qu et al., 2023). The 3D information of the atmosphere based on measurements of these instruments (Mason et al., 2023) will be used as input for 1D and 3D-radiative transfer calculations (Cole et al., 2022). The calculated top-of-atmosphere (TOA) solar and thermal fluxes leaving Earth to space are finally compared to the short and longwave fluxes, estimated from radiance measurements from the broad-band radiometer (BBR) to achieve radiative closure (Tornow et al., 2015; Barker et al., 2024). Various terms
exist in the literature to refer to radiative fluxes, such as flux density and irradiance. Although these terms are often used interchangeably within the scientific community, with some debate, they all refer to the same quantity.

The BBR aboard EarthCARE will measure accurate shortwave (SW) and total-wave (TW) radiances in a push broom along-track configuration at three fixed viewing angles (nadir, 55° fore, and 55° aft) (Caldwell et al., 2017). The telescopes measure TW radiances from 0.25 μm to beyond 50 μm. The shortwave channel covers radiances from 0.25 μm to 4 μm by applying
an uncoated synthetic quartz filter mounted on a rotating drum to the telescopes. Synthetic longwave (LW) radiances are obtained by subtracting the SW from the TW channel. The SW-LW inter-channel contamination and the instrument spectral response effects are removed in the BM-RAD processor (Velázquez Blázquez et al., 2023). The obtained "unfiltered" SW and LW radiances are then used as input for the BMA-FLX processor, described hereafter to estimate top-of-atmosphere fluxes. BM-RAD reports radiances for the three viewing angles at fixed spatial resolutions, being the "standard resolution" of 10km
x 10km, and the configurable "assessment domain resolution" of 21 km along-track by 5 km across-track employed in the radiative closure (Barker et al., 2024).

Due to the highly anisotropic character of some physical phenomena, like the reflection of solar radiation by clouds, the estimation of the radiative fluxes from measured radiances at a single Sun-observer geometry is challenging. In recent years, various approaches have been developed to estimate the anisotropy of the observed scenes using the so-called angular distribu-
tion models (ADM). An overview of different approaches is described in Gristey et al. (2021).

Several studies have analyzed different radiance-to-flux approaches for the BBR over the past years of EarthCARE devel-opment (Domenech et al., 2011a; Domenech et al., 2011b; Domenech and Wehr, 2011; Tornow et al., 2019, 2020, 2021). This paper presents the final implementation of the mission's operational flux retrieval algorithm. The BMA-FLX SW algorithm is scene dependent and has been constructed from six years of Clouds and the Earth's Radiant Energy System (CERES) and
Moderate Resolution Imaging Spectroradiometer (MODIS) Terra and Aqua measurements using an artificial neural network approach. The BMA-FLX LW algorithm follows the approach described in Clerbaux et al. (2003a), and is based on corre-

lations between BBR radiance anisotropy and the spectral information provided by the EarthCARE's MSI radiances derived from radiative transfer calculations (RTC). As the processor name indicates, inputs from BBR (radiances), MSI (radiances, cloud top height, and cloud fraction) and ATLID (cloud top height) are used to estimate fluxes.

Providing instantaneous thermal and solar flux estimates, BMA-FLX allows a final comparison with calculated TOA fluxes using radiative transfer models and evaluates if the mission goal of $10~\mathrm{Wm}^{-2}$ has been reached (Wehr et al., 2023). The accuracy requirement placed under the EarthCARE radiative closure's goal is defined in the EarthCARE mission Requirements Document (MRD) (Wehr, 2006).

## 2   Algorithm description

Building on the experience gained from previous Earth Radiation Budget missions such as CERES (Wielicki et al., 1996), GERB (Harries et al., 2005), and ScaRaB (Kandel et al., 1998), an angular distribution model (ADM) methodology is considered an appropriate candidate to meet EarthCARE's radiative requirements.

The solar flux leaving the Earth-atmosphere system is obtained by integrating the radiance field, $L(\theta_0, \theta, \phi)$, over the solar zenith angle (SZA, $\theta_0$), the viewing zenith angle (VZA, $\theta$), and the relative azimuth angle between the Sun and the satellite 70   view (RAA, $\phi$) as follows:

$$F(\theta_0) = \int_0^{2\pi} \int_0^{\pi/2} L(\theta_0, \theta, \phi) cos(\theta) sin(\theta) d\theta d\phi. \tag{1}$$

The thermal flux has the same form but without the dependence on the solar zenith angle, $\theta_0$, and with $\phi$ denoting the viewing azimuth rather than the relative azimuth.

To estimate the instantaneous top-of-atmosphere radiative flux from a single radiance measurement, some information of the 75   angular variation of the radiation field that constitutes the flux is needed. A primary error in deriving flux arises from the lack of knowledge of the target's anisotropy. The ADM's methodology have accurately represented this variation, e.g., (Su et al., 2015), and so are used as the basis for flux retrieval.

An ADM represents an estimate of the radiation field anisotropy and has been used to derive the anisotropic factor ($R$), which is the ratio between the equivalent Lambertian flux and the actual flux and enables the conversion of a single radiance 80   measurement at a given angle to radiative flux:

$$R(\theta_0, \theta, \phi) = \frac{\pi L(\theta_0, \theta, \phi)}{F(\theta_0)}. \tag{2}$$

The SW flux retrieval algorithm is defined independently from the LW design since the anisotropy of the radiance field in the SW and LW regimes depends physically on different geophysical parameters.

The BBR instrument observes each target on Earth from three different directions almost simultaneously (about 3 minutes 85   between the fore and aft views), providing information on scene anisotropy. However, the BBR's multi-pointing capability is not directly utilized in the flux retrieval. Previous studies have analyzed the advantage of using the BBR's multi-angle views

to directly constrain the radiance-to-flux conversion (Domenech and Wehr, 2011; Domenech et al., 2011a), and this will be explored in later iterations of the processor. Currently, the operational processor aligns with the contingency requirements of the mission, which mandate that flux estimates must be retrievable from each individual telescope to address potential critical failures. As a result, the radiances acquired by the three telescopes for a single scene are not converted into a flux estimate. Instead, different ADMs are applied to the radiances, resulting in three separate flux estimates for the observed scene. As the outgoing flux is only dependent on the solar geometry and the radiometric properties of the atmospheric-surface domain, the three fluxes estimated by the ADMs should result in a similar flux assuming perfect instrument and retrieval responses. This is not the case in a real scenario, where discrepancies in ADM-based fluxes obtained from different viewing geometries for the same surface-atmosphere scene can still be significant (Domenech et al., 2012). In the BMA-FLX processor, the LW ADMs are assumed to be a function of only $\theta$, while the SW ADMs depend on $\theta_0$, $\theta$, and $\phi$.

## 2.1 BMA-FLX SW algorithm

### 2.1.1 SW radiance-to-flux description

Domenech et al. (2011b) developed angular models for the BBR from Monte-Carlo RT simulations to construct a synthetic ADM specifically defined for the multi-pointing capability of the BBR. Two main conclusions were extracted: methodologies entirely based on simulated data are biased and difficult to extrapolate for real use; and multi-view flux conversion algorithms improve performance of single-view ADMs in highly anisotropic scenes (Domenech et al., 2012).

The artificial neural network (ANN) method described by Loukachine and Loeb (2003, 2004) and adopted for ScaRaB-3 (Viollier et al., 2009) using CERES data to train the models, was modified to incorporate simultaneous use of along-track radiance measurements in Domenech and Wehr (2011); Tornow et al. (2019). These studies are further developed in the BMA-FLX SW processor, where the radiance-to-flux conversion algorithm employs a feed-forward backpropagation ANN to model the ADM-based fluxes from the CERES Single Scanner Footprint TOA/Surface Fluxes and Clouds (SSF) Edition 4 product (Loeb et al., 2005; Su et al., 2015). Backpropagation is the generalization of the Widrow-Hoff learning rule (Widrow and Lehr, 1990) to multiple-layer networks. Input vectors and the corresponding target networks are used to train a network until it can approximate a function that associates the input vectors with specific output vectors. The backpropagation employed here is a gradient descent algorithm in which the network weights are moved along the negative of the performance function gradient.

The BMA-FLX processor retrieves fluxes for each BBR telescope, checks the consistency of the estimates, and combines them according to their estimated errors, $\varepsilon_F$, and the corresponding radiance error provided by the BM-RAD processor, $\varepsilon_R$.

### 2.1.2 Scene definition

Scene definition refers to the classification of targets based on surface and cloud properties, as well as the angular geometry that defines the ADM model used for BBR observation. During the ADM development, the stratification of the scene definition, scene classes, determines the number of anisotropic models, which in turn dictates the number of datasets required for training the networks that construct the ADM.

Our scene classes consists of seven static surface types (ocean, forest, savanna, grassland, shrub, desert/ bare soil, and
permanent snow), plus two dynamic ones (fresh snow and sea ice), and four cloud fractions, CF (cloud-free, partly covered
$0.1 < CF < 50$, mostly covered $50 \leq CF < 99$, and overcast), taking into account that for overcast conditions the categories
forests, savannas, grasslands, shrubs, and desert are grouped in a new category named land. This classification is applied to
each BBR view considering the corresponding scattering regime in the oblique telescopes (forward and backward). The scene
is defined separately for forward and backward scattering directions because the angular geometry significantly impacts the
observed radiative properties of the target. This detailed classification is necessary because the scattering characteristics of a
scene can vary significantly between forward and backward directions. By accounting for these variations, the ADM can more
accurately model and predict the radiative properties of different scenes. This approach ensures that the ADM is trained on a
comprehensive set of datasets, enhancing its reliability in various observational conditions. Table 1 presents the classification of
scenes, resulting in a total number of 96 scenes considered for the ADM. Cloud fractions retrieved from the CERES/MODIS
algorithm and based on MODIS pixel-level measurements (Minnis et al., 2021) are used in the training datasets. The static
categories considered for training are obtained from the International Geosphere–Biosphere Programme (IGBP) land cover,
while the fresh snow and sea ice surface types are derived from the CERES SSF snow information as a combination of the
microwave snow/ice map from the National Snow and Ice Data Center (NSIDC) and the snow/ice map from the National
Environmental Satellite, Data and Information Service (NESDIS).

| Scene definition | |
|---|---|
| Surface type | Sky condition |
| Land | Overcast |
| Forest<br>Savanna<br>Grassland<br>Shrub<br>Desert/ Bare soil | Clear-sky, partly covered, mostly covered |
| Ocean<br>Fresh snow<br>Permanent snow<br>Sea ice | Clear-sky, partly covered, mostly covered, overcast |

**Table 1.** ADM scene definition based on a macrophysical cloud description and surface type adopted in the BBR SW algorithm. Nadir and both scattering regimes (forward and backward) apply to each scene class.

### 2.1.3 Model training

A geophysical database for model training has been created using six years of Level 2 CERES radiance and flux measurements. This database optimizes angular geometry variety and matching with the imager. The data includes measurements from the CERES FM1 and FM2 instruments on the Terra satellite and the FM3 and FM4 instruments on the Aqua satellite, all in Fixed Azimuth Plane Scan (FAPS) mode, collected between 2000 and 2005. Additionally, it includes data from the CERES FM1 and FM3 instruments in cross-track scan mode for the year 2007.. For cloudy scenes, MODIS radiances and cloud masks matching the CERES measurements are collected to study the presence of clouds and their implication. Additionally, climatology data on surface albedo and aerosol optical depth, along with meteorological reanalysis data on atmospheric gases, vegetation, and wind speed for cloud-free scenes, are gathered to analyze the amount of radiation reflected by the surface, the extinction of SW radiation by dust and haze, the absorption/emission of radiation by atmospheric gases, the role of vegetation in the interaction between Earth's surface and atmosphere, and the vertical wind changes, respectively.

The geophysical database is designed to match the scene definition. For each scene class, pairs of input and output vectors are selected as training datasets. The output values represent CERES anisotropic factors, while the input vectors are chosen from a predefined list of parameters that characterize scene anisotropy (Tab. 2). These parameters may or may not effectively characterize anisotropy for each scene class. To determine the most significant input parameters for each class and scattering direction, we assess the variables' importance in reproducing the anisotropic factors. This assessment is conducted using two independent tests: a Random Forest regression-based permutation test and a Genetic Algorithm applied to a Linear Model, both of which largely agree on the optimal subsets of input parameters. Subsets derived from the Random Forest test produced slightly better performance in ANN-based flux prediction, and, therefore, represent final subsets of ANN inputs. SW radiances at the nadir, fore, and aft BBR viewing directions and the illumination/viewing geometry (SZA, VZA, RAA) from CERES are consistently included as inputs. Since SW ADMs are developed separately for each scattering direction, the CERES observations are matched with either forward or backward scattering directions based on the relative azimuth angle during the input selection process.

To consider the anisotropic characteristics over cloud-free land surfaces, we use RossThick-LiSparse geometric, volumetric, and isotropic bidirectional reflectance factor (BRDF) parameters, used to define the albedo depending on the presence of direct and diffuse components and taken from an albedo climatology derived from the MODIS MOD43B product (Qu et al., 2022). Additionally, we employ the leaf-area index (LAI) for high and low vegetation, and the aerodynamic surface roughness length from meteorological reanalysis (Poli et al., 2016). We also consider the hotspot effect as approximated in Rahman et al. (1993) as input to describe the enhanced reflectivity of the surface for an observational geometry close to the solar illumination geometry. Over ocean, aerosol optical depth (AOD) and wind-speed are chosen as parameters for the network training. AOD is based on an AeroCom climatology (Koffi et al., 2016), and wind-speed from meteorological reanalysis (Poli et al., 2016). In addition, the glitter radiance normalized by the incident irradiance (sun-glint reflectance), given by Jackson and Alpers (2010), is also used in the training to represent radiance reflected by the ocean surface with the same angle as the satellite

viewing angle. To further consider anisotropy changes due to atmospheric characteristics, we also included the total-column water-vapour and total-column ozone from reanalysis. Snow and ice surfaces are assessed independently.

The inputs for creating training sets for cloud scenes include cloud cover and radiances from the MODIS bands closest to those of MSI. The underlying assumption is that the non-linear combination of narrow-band radiances provides adequate information about the anisotropy of cloudy scenes, eliminating the need for using imager-retrieved cloud properties. Imager radiances are analyzed separately over cloud-free and cloudy parts of the observed scene. The optimal combination of narrow-band radiances includes the 0.67, 0.865, 10.8, and 12.0 $\mu$m MSI bands. This selection was primarily influenced by the avail-

ability of bands with similar central wavelengths in the MODIS PSF-weighted radiances provided in the CERES SSF products and by their importance in retrieving cloud properties using the MSI (Hünerbein et al., 2024, 2023). While the short-wave infrared MSI bands significantly impact ADM performance, the CERES Ed4 product's PSF-weighted imager statistics for these MODIS bands did not include statistics for clear and cloudy areas over the CERES footprint, which are essential for constructing the BBR ADMs.

Each training dataset is employed to train a network. Since the networks depend on the bias/weight initialization, a Nguyen-Widrow initialization method is used to create additional models for each scene class. Resulting ANN models are crosschecked against a validation dataset (20% of the training data) and the networks with better performance are selected for each scene class of the ADM. After training, the resulting networks are evaluated against a CERES validation dataset. The root mean square errors (RMSE) of the ANN-based flux estimates compared to the original CERES derived fluxes define the theoretical

uncertainties for each scene class.

| SW ADM inputs | |
|---|---|
| MSI bands (0.67, 0.865, 10.8, 12.0 $\mu m$) | Wind speed |
| Cloud cover | Surface roughness |
| Viewing geometry (SZA, VZA, RAA) | Atmospheric gases (total-columns water-vapour and ozone) |
| SW radiances at the BBR viewing directions | AOD |
| Surface albedo | Sunglint reflectance |
| LAI for low and high vegetation | Hotspot effect |

**Table 2.** Model inputs retrieved from a geophysical database for model training using six years of Level 2 CERES data. Climatology and meteorological reanalysis data for both cloudy and clear-sky scenes.

### 2.1.4 Scene identification

The scene identification is the first step in the flux retrieval. The illumination and viewing geometry of the scene classify the fore and aft BBR observations into either the forward or backward scattering regime based on their relative azimuth angle.

Surface types at a 1-km resolution are collected from a simplified version of the Global Land Cover Characterization (GLCC) dataset (Belward and Loveland, 1996) at the BBR pixel level, and the predominant surface type is selected for cloudy scenes. For clear-sky cases, which may involve a combination of surface types, the observation is classified as a mixed scene if more than one surface type is present. The retrieval algorithm then uses the two surface types with highest coverage within the BBR pixel. Two ADM scene classes are selected, obtaining two SW anisotropic factors for the footprint. The pure anisotropic factors obtained for primary and secondary surface areas within the BBR fore, aft, and nadir observations are combined into a mixed anisotropic factor (Bertrand et al., 2005), weighting the pure anisotropic factors by their respective (scaled) surface coverages and TOA albedos. The mixed flux is then calculated using the corresponding SW radiance and the mixed anisotropic factor. The dynamic surface types, sea ice and fresh snow, override the GLCC selection, and are obtained from the European Centre for Medium-Range Weather Forecasts (ECMWF) high-resolution forecasts contained in the X-MET product (Eisinger et al., 2024).

In clear-sky conditions, without cloud parallax and at sea-level locations, the default surface co-registration is used. However, if a new co-registration is needed (i.e., surface elevation is greater than 0 and/or a cloud is observed in the nadir view), the ADM scene classes for the BBR oblique observations are reconstructed. When the digital elevation model (DEM, Berry et al. (2010)) indicates elevations above 0, the scene identified for the nadir view is also used in the BBR oblique views in the new co-registration. To reconstruct the scene classes for the oblique views in cloudy nadir conditions, the cloud fraction observed in the nadir view and the surface definition from the displaced oblique views are used (new fore and aft BBR views intersecting the cloud top observed in the nadir view, see section 2.1.5). The cloud mask and cloud top height, used to match the views, are derived from the MSI operational retrievals (Hünerbein et al., 2023).

MSI can separate snow from most obscuring clouds, but it does not consistently discern optically thin cirrus clouds from snow. Snow and ice surfaces not only share similar reflectance and brightness temperatures with the overlying clouds but can also be colder than clouds themselves due to inversions in the atmospheric temperature profile (Liu and Key, 2003). Techniques commonly used to detect inversions cannot be employed with the MSI due to the lack of water vapor absorption bands (6.7 and 7.2 μmm) and the carbon dioxide band (13.3 μmm). Instead, the cloud top height derived from the EarthCARE's LiDAR (ATLID) is used to verify the cloud top height retrieved by MSI over cold surfaces. In cases where there is inconsistency between the CTHs provided by ATLID and MSI over cold surfaces, ATLID prevails. The cloud cover and CTH used in the processing are adjusted accordingly.

### 2.1.5 Determination of the solar flux at reference level

Once the scene is identified, the anisotropic factors required to convert BBR radiances into fluxes are selected accordingly. Given that cloud parallax can affect this selection—and consequently the flux calculation—the radiance-to-flux procedure is integrated with a process of radiance co-registration.

BBR's three telescopes are oriented towards nadir and $50°$ fore and aft along the ground-track. Projecting the oblique views to Earth's surface results in viewing zenith angles around $55°$. BBR radiance measurements are typically co-registered at surface level by default. In clear-sky conditions, the primary emission or reflection observed is from the surface, making this default radiance collocation adequate, provided there is no significant elevation. However, in the presence of clouds, the most radiatively significant reflection or emission occurs in a layer between the surface and the top of the cloud, not necessarily at the surface. Therefore, to ensure accurate flux estimates from the three BBR radiances for each BBR sample, these measurements must be co-registered to this vertical layer, known as the reference level altitude. Without this co-registration, the flux estimates cannot be accurately compared.

To accurately determine the co-registration for the aft and fore views, the nadir view is used as the reference. Fore and aft radiances are matched to the nadir sample at a reference level altitude. Flux measurements for nadir and fore observations across several orbits of the Advanced Along-Track Scanning Radiometer (AATSR) were calculated using a radiance narrow-to-broadband conversion. The SW NB to BB used the AATSR solar channels 0.6, 0.8 and 1.6 $\mu$m (Clerbaux et al., 2005), while the LW used the thermal channels 10.8 and 12.0 $\mu$m in a second order regression. This approach was employed to identify the optimal reference level height for both SW and LW radiances. The findings indicated that using a variable reference level is essential to fully exploit the multi-angular viewing capabilities of the BBR in the flux retrieval algorithm. The reference level algorithm was tested in a 3D environment generated with a Monte Carlo radiative transfer code, providing satisfactory results as reported in Barker et al. (2024).

In the SW regime, co-registering BBR radiances using brightness temperature (BT)-derived CTH results in higher errors compared to the default surface co-registration. A more effective approach involves calculating the fluxes for oblique BBR measurements in 1-kilometer increments, i.e., BBR sampling. These calculations consider the atmospheric path of each radiance that intersects the vertical levels between the surface and the 90th percentile CTH derived from MSI data. The reference level, $H_{sw}$, is then selected by minimizing the flux differences between nadir, fore, and aft fluxes as follows:

$$H_{sw} = \arg_x \min \left( \left| F^x_{sw,\text{fore}} - F^x_{sw,\text{aft}} \right| + \left| F^x_{sw,\text{fore}} - F^x_{sw,\text{nad}} \right| + \left| F^x_{sw,\text{nad}} - F^x_{sw,\text{aft}} \right| \right), \tag{3}$$

where $x$ represents the vertical layers from the surface to the cloud top height, and $F^x_{sw,\text{nad}}$, $F^x_{sw,\text{fore}}$, and $F^x_{sw,\text{aft}}$ are the nadir, fore, and aft SW fluxes calculated for each layer. The new co-registration of the oblique views is given by $d = x \tan(\theta_{obl})$ for the $x$ value that minimizes the flux differences.

The fore, aft, and nadir SW flux estimates, $F_{sw,j}^i$, are calculated from the anisotropic factors, $R_{sw,j}^i$, using the following equation:

$$\bar{F}_{sw,j}^i(\theta_0) = \frac{\pi L_{sw,j}^i(\theta_0, \phi)}{R_{sw,j}^i(\theta_0, \phi)}, \tag{4}$$

where the index $i$ represents the nadir, fore, and aft views, and the index $j$ represents the observed scene.

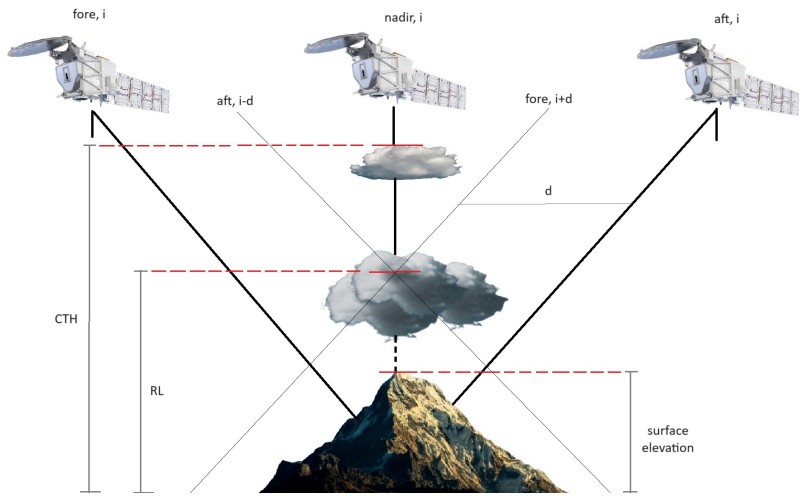

**Figure 1.** Diagram illustrating the co-registration applied to SW radiance measurements in a two-layer cloud system. Satellite image provided courtesy of ESA.

Figure 1 shows an example of co-registration of fore and aft SW measurements for a scene with two cloud layers. The optically thin high cloud is semi-transparent to SW radiation, with the main reflection occurring at the core of the lower thick cloud. While the CTH derived from MSI would set the reference level, RL, at the top of the high cloud, the minimization technique redefines the reference level to the center of the lower thick cloud. The oblique views that minimize the flux differences are displaced by a distance $d$ from the default surface co-registration. These oblique observations, which are expected to be
reflected from the same atmospheric region, are then co-registered with the nadir observation of sample $i$.

### 2.1.6   Combining resulting SW fluxes

To provide the most reliable flux estimate for the target observed by the BBR telescopes during operations, the three fluxes retrieved from the radiances co-registered at $H_{sw}$ undergo an internal consistency check to assess discrepancies between them. These fluxes are then merged, incorporating the uncertainties from the ADM construction and the observed deviations between
the fluxes.

     The presence of clouds along the optical path creates inconsistencies between the atmospheric domains observed by nadir and oblique views (see Figure 2). Therefore, the cloud presence must be checked before combining the different fluxes. The

MSI CTH is employed to detect either clouds in the surroundings of the nadir clear-sky domain or higher clouds than those observed in the nadir domain. Given a tropopause height, $h_T$, the oblique observation is considered affected by cloud parallax if the CTH of clouds along the optical path is located at a height between the surface elevation of the nadir clear-sky domain (or CTH of clouds of a cloudy nadir domain) and the tropopause, the cloud height being $h_i = \frac{d_i}{\tan(\theta_{obl})}$, where $d_i$ ranges from 1 to $d_{maxparallax} = h_T \tan(\theta_{obl})$ and $\theta_{obl}$ is the viewing zenith angle of the oblique telescopes.

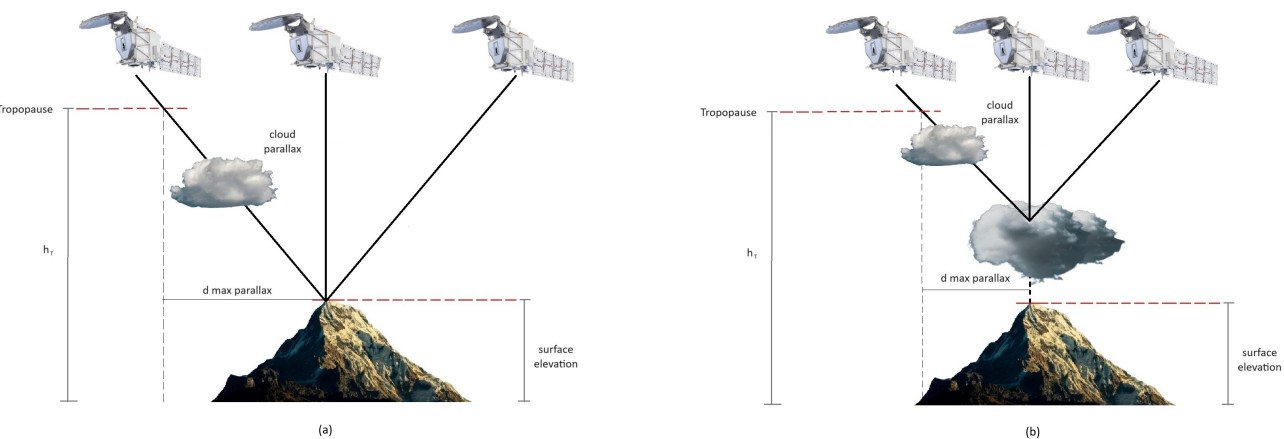

**Figure 2.** Diagram illustrating the presence of cloud parallax in a co-registered fore observation under nadir-observed clear (left panel) and cloudy sky conditions (right panel). Satellite image provided courtesy of ESA.

In absence of parallax, the discrepancies between the fluxes derived from each combination of nadir, aft, and fore observations are calculated as follows

$$\Delta_{y-z} = 100 \frac{|F_y - F_z|}{\frac{1}{2}(F_y + F_z)}, \tag{5}$$

where $\Delta_{y-z}$ is the fractional error (0-200%) between the SW fluxes obtained from BBR views $y$ and $z$.

The BBR ADM evaluation was carried out using CERES flux estimates from the database described in Section 2.1.3. The BBR ADMs simulate CERES radiative flux retrievals, which are considered the "truth" in this analysis. The evaluation dataset is a randomly selected subset of the initially identified training data, deliberately excluded from the training process.This validation test demonstrated low RMS errors and no significant bias, highlighting the accuracy of the model's performance.

Following the strategy used in the training, the input parameters (NB radiances, BB radiances, cloud fraction, and surface parameters) are obtained from the CERES SSF and auxiliary products. For each ADM scene class, a flux estimate was obtained for every CERES radiance measurement, and the RMSE was computed for the entire bin. The RMSE for each scene class represents the performance of the BMA-FLX SW model compared to CERES, indicating the minimum uncertainty associated with the ADMs.

The ADM uncertainties and the flux discrepancies between BBR views are used to combine the derived fluxes for each view $i$ as follows

$$\bar{F}_{sw} = \left[ \sum_{i=1}^{3} \frac{\delta^i}{\varepsilon^i_{F_{sw}} \pi \varepsilon^i_{L_{sw}}} \right] \left[ \sum_{i=1}^{3} \frac{\delta^i F^i_{sw}}{\varepsilon^i_{F_{sw}} \pi \varepsilon^i_{L_{sw}}} \right], \tag{6}$$

with the weights defined by the uncertainties. $\varepsilon^i_{F_{sw}}$ are the flux uncertainties arising from the ADM, and $\varepsilon^i_{L_{sw}}$ the unfiltered
radiance errors provided by the BM-RAD processor (Velázquez Blázquez et al., 2023).

When all $F^i_{sw}$ show discrepancies < 10%, $\delta^i = 1$ for all $i$. When two $F^i_{sw}$ agree to within $\pm 10\%$, $\delta^i = 1$ with the outlier getting $\delta^i = 0$. If all $F^i_{sw}$ show fractional errors > 10%, only the lowest $\varepsilon^i_{F_{sw}} \pi \varepsilon^i_{L_{sw}}$ uses $\delta^i = 1$, and $\delta^i = 0$ for the others.

## 2.2 BMA-FLX LW algorithm

### 2.2.1 LW radiance-to-flux description

The methodology proposed is based on Stubenrauch et al. (1993), Domenech et al. (2011b) and on the operational GERB LW flux estimation (Clerbaux et al., 2003a, b). The algorithm estimates the anisotropy as a function of the MSI thermal channels information through polynomial second order regressions on the MSI channels brightness temperatures.

Previous studies for GERB (Dewitte et al., 2008; Clerbaux et al., 2009) have shown that using a single multi-spectral regression for all scenes can cause large biases for semi-transparent clouds. This problem was also highlighted in the early stages of the selection and validation of the BMA-FLX processor algorithm. To overcome this problem, it was proposed to use a set of different regressions, instead of one, in line with the approach adopted by the EUMETSAT Central application facility (CAF) for the SEVIRI Outgoing Longwave Radiation (OLR) estimation (EUTMETSAT, 2010). Although there is no water vapour channel in the MSI, the use of the infrared channels difference overcomes this lack. To reduce the instability due to the collinearity between the MSI radiances, the model is constructed using as predictors the brightness temperatures in the 10.8 μm channel and the differences between brightness temperatures ($T_b$) in the 10.8 μm and 12.0 μm channels. Therefore, the predictor variables have been defined as $z_1 = T_b (10.8 \text{ μm})$ and $z_2 = T_b(12.0 \text{ μm}) - T_b (10.8 \text{ μm})$ and $a_i$ are the coefficients of the regression, dependent on the viewing zenith angle $\theta$.

$$R(\theta) = a_0 + a_1 z_1 + a_2 z_2 + a_3 z_1^2 + a_4 z_1 z_2 + a_5 z_2^2. \tag{7}$$

Different regressions, and consequently different anisotropy models, have been developed for thermal radiances in bins of 20 $\text{Wm}^{-2}\text{sr}^{-1}$ and also every 5 degrees in VZA. The lower anisotropy in the LW domain, with lower errors expected in the inversion process (Bodas, 2002), enables sufficiently accurate anisotropy models to be derived from radiative transfer (RT) simulations. Although some studies (Minnis and Khaiyer (2000) and Minnis et al. (2004)) have shown that under certain situations longwave radiances can strongly depend on the azimuth angle, our simulations have assumed azimuthal symmetry of the longwave radiation. The theoretical anisotropic factors, $R(\theta)$, have been estimated from theoretical simulated thermal radiances and fluxes computed using LibRadtran 1.4 (Mayer and Kylling, 2005) and SBDART (Ricchiazzi et al., 1998) radiative transfer models, following the standard approach, adapted from Eq. 2.

A detailed description of the radiative transfer databases used in both unfiltering of the BBR and the LW radiance-to-flux conversion can be found in Velazquez et al. (2010).

The proposed methodology using level-1 brightness temperatures from the M-RGR product (Eisinger et al., 2024) as predictors has the advantage of not using a bin classification for the estimation of the flux, avoiding potential errors due to a misidentification of the scene.

In summary, the LW flux retrieval algorithm provides instantaneous TOA thermal radiative flux estimates for the BBR measurements co-registered at the reference level, converting the broadband radiance measurements (4 μm - 500 μm) into flux estimates. It is assumed that the combination of the off-nadir and nadir BBR fluxes improves the estimation of the flux, and the correlation between the broadband radiances and the spectral signature for the radiation field can be exploited to reduce the LW flux error.

### 2.2.2 Combining resulting LW fluxes

The three BBR thermal unfiltered radiances use the default surface co-registration for clear sky scenes but in presence of clouds are co-registered at a reference level defined by the percentile 90th of the MSI CTH from the M-COP product (Hünerbein et al., 2024). The only difference between the SW and LW domains regarding the co-registration of the views is the altitude and definition of the reference level for the co-registration of the off-nadir views.

In a standard limb-darkening situation, an oblique BBR observation, obtained at a VZA close to 55 degrees, should lead to a highly accurate flux estimation (Bodas-Salcedo et al., 2003; Smith et al., 1994). Thus plane parallel assumptions would indicate that greater weight should be placed on the fore and aft views. However, for non plane-parallel scenes, the average of the three views maximizes the sampling of the radiance field, reducing the error on the flux estimation.

The approach followed in the LW regime consists in using the weighted average of the LW fluxes obtained from the three BBR radiances to optimize the use of the nadir view:

$$F = \frac{(1-\alpha)\pi L_{fore}}{2R_{fore}} + \frac{(1-\alpha)\pi L_{aft}}{2R_{aft}} + \alpha \frac{\pi L_{nad}}{R_{nad}}, \tag{8}$$

where the anisotropic factors are obtained from the NB observations of the MSI and $\alpha$ is the weighting factor of the nadir view. The latter is important to discriminate the scene types where the LW flux accuracy is limited by the angular sampling. Note that as in the SW case, views affected by parallax will not be used in the combined flux calculation.

The best value for the $\alpha$ parameter was evaluated in a previous study using all the available CERES FM2 true along-track data (6 days of February 2005). In this study, a flux from the Direct Integration (DI) of LW radiances along the track has been calculated and compared to a linear combination of the CERES fluxes for each BBR view (fore, nadir, aft). The best linear combination of the 3 views resulted in weighting factors very similar for the fore and aft views and slightly lower for the nadir view being $\alpha$ very close to 1/3. This means that in the real world the thermal flux is on average more dependent on 3-dimensional effects than on plane parallel ones. Therefore the current version of the BMA-FLX thermal flux retrieval algorithm assigns the same weight to each of the 3 views. A methodology to discriminate scenes with standard limb-darkening

behaviour is expected to improve this approach and this will be further consolidated using real EarthCARE data during the commissioning.

## 2.3 Convolution of MSI radiances and cloud properties

MSI radiances, brightness temperatures, and cloud retrievals (i.e., cloud fraction and CTH), which serve as input variables for flux retrieval models, are averaged over the different BBR resolutions (standard, small, full and assessment). This averaging process must account for the signal response characteristics of the BBR instrument to ensure a consistent comparison between the averaged imager values and the energy measured by the BBR.

MSI retrievals are integrated over the BBR sample using the instrument's point-spread function (PSF). Specifically, solar and thermal radiances, cloud mask, and cloud top height (used in the reference level height) at MSI pixel level are convolved within the BBR sample to produce PSF-weighted mean and standard deviation at the BBR resolution. These values are then used as inputs in the BBR SW and LW ADMs.

Let $x$ represent a general input parameter over the 95% energy of the BBR sample. The weighted average value of $x$ is calculated as:

$$\bar{x} = \frac{\sum_i \omega_i x_i}{\sum_i \omega_i},\tag{9}$$

where the index $i$ denotes each position within the BBR sample, and the $\omega_i$ represents the PSF weight at position $i$. The denominator in Eq. 9 is the sum of the PSF weights over the sample, which should approximate 95% of the total measured energy.

## 3 Evaluation of the BMA-FLX processor

To assess the robustness of the BMA-FLX processor, we implemented a two-step evaluation process. The primary objective is to evaluate the expected accuracy and reliability of the BMA-FLX processor in retrieving radiative fluxes using data from simulated EarthCARE orbits. Full validation of the fluxes retrieved by the BMA-FLX processor is anticipated during EarthCARE's Commissioning Phase. However, these two steps are designed to isolate and analyze different sources of uncertainty and error in the flux retrieval process. Section 3.2 describes the processor uncertainty assessment using model truth cloud profiles and model truth snow and sea ice properties. This step evaluates the intrinsic performance of the BMA-FLX processor by eliminating uncertainties introduced by Level 2 cloud retrieval algorithms from other instruments. Here, input data derived from MSI and ATLID's forward models are replaced with data directly taken from geospatial simulations. By using model truth cloud and snow information, this assessment focuses on the processor's ability to handle ideal input conditions without compounded errors from preceding algorithms. Section 3.3 presents an assessment of end-to-end uncertainty using operational L2 products. This step evaluates the performance of the BMA-FLX processor under realistic conditions, where input data include the uncertainties and errors from the operational L2 EarthCARE processors. The BMA-FLX processor ingests L2 products derived

from simulated L1 data, simulating an operational scenario. This provides insights into the processor's anticipated issues and robustness in the presence of non-ideal inputs.

The isolation of processor performance using model truth data allows the identification and resolution of intrinsic issues without external retrieval errors. In contrast, the realistic operational conditions incorporate all sources of uncertainties from the operational L2 products, providing a comprehensive evaluation of the processor's performance in simulated mission conditions. The main limitations of this evaluation include reduced scene diversity and potential discrepancies between the radiative transfer simulations used in the simulated geophysical data and the modeled EarthCARE products. It is also important to note that the radiative transfer simulations are based on a plane-parallel model, which complicates the comparison between the model's true flux and the fluxes derived from our empirical models. The test frames are based on three specific EarthCARE orbits, which, although diverse, may not encompass all possible atmospheric and surface conditions encountered globally. Additionally, discrepancies in surface definitions between the RTC model and the BMA-FLX processor can affect flux retrieval accuracy, particularly in clear sky conditions.

## 3.1 EarthCARE test frames

The EarthCARE mission features a simulator multi-instrument framework (ECSIM) (level-0 to level-2b) to test operational algorithms and validate the entire product processing chain. The BMA-FLX processor interfaces, within EarthCARE processing scheme, with M-CLD to receive the MSI-based CTH and cloud mask from M-COP and M-CM, to M-RGR to receive MSI regridded radiances and brightness temperatures, to A-LAY to receive the ATLID-based cloud top height from A-CTH, to BM-RAD to receive unfiltered radiances, to X-MET to receive high-resolution forecasts, and to ACM-RT (Cole et al., 2023) to provide co-registered radiances and fluxes.

3D atmosphere-surface data is produced by the Global Environmental Multi-scale (GEM) Numerical Weather Prediction (NWP) model for three EarthCARE frames (1/8 of an EarthCARE orbit) (Qu et al., 2022). Donovan et al. (2023) describes the radiative-transfer models and instrument models to prepare simulated Level 1 data for testing L2 EarthCARE processor interfaces and evaluating processor performances in these test frames. Level 2 data derived from the EarthCARE forward processors and the RTC radiative fluxes are available for testing BMA-FLX in the test frames. The L1 simulated data and model truth fields for three GEM-derived scenes are available from https://doi.org/10.5281/zenodo.7117115 (van Zadelhoff et al., 2023a).

The test frames correspond to orbits over Halifax (Canada), Baja California and Hawaii (USA). In the high-latitude portion of the Halifax scene, nighttime mixed-phase clouds transition from deeper formations, featuring supercooled liquid, to mixed-phase clouds. Further south, the clouds become more broken and shallower. Near the center, a storm system with supercooled layers, convective precipitation, and ice clouds is present, followed by cloud-free areas and shallow, low-altitude water clouds. The Baja scene features significant topographical variation compared to the Halifax scene and contains large regions of thinly distributed aerosols. High-level ice clouds are present south of $35°$ N. Near the center of the scene, above the Rocky Mountains, there are extensive regions of optically thick ice and water clouds. The Hawaii scene is almost entirely over ocean. The nadir

track is completely over ocean, with a few of the smaller Hawaiian islands appearing within the MSI track. This scene includes areas of clear sky, upper-level cirrus clouds, and a tropical convective system near the center of the scene.

## 3.2 Processor uncertainty assessment

The errors in the L2 retrieval algorithms from ATLID (A-CTH), MSI (M-RGR, M-CM and M-COP) and BBR (BM-RAD), together with the meteorological data from X-MET, introduce uncertainties that impact both the solar and thermal fluxes retrieved using ADMs in the BMA-FLX processor. Before evaluating the performance of the BMA-FLX algorithm and the accuracy expected in the retrieval, it is necessary to get rid of the uncertainties in the data inputs. The analysis of how the processor deals with non-ideal inputs will be addressed in section 3.3.

The inputs for BMA-FLX derived from the EarthCARE forward models are replaced by data directly taken from geospatial simulations, used as inputs for the L1 processors. New inputs are generated using the model truth cloud profiles and snow/ice conditions, thus avoiding the errors introduced by the other algorithms. The starting point was the extinction profiles at 680 nm obtained from the Global Environmental Multi-scale (GEM) Numerical Weather Prediction (NWP) model (Qu et al., 2022), and the surface properties from Donovan et al. (2023) . The cloud optical thickness (COT) of the three-dimensional scene is

regridded and resampled to the MSI grid. Different COT thresholds were considered to calculate both the cloud mask and the cloud top height, 0.1 being found to have the best metrics in the analysis of the flux results. Once the MSI-like model truth cloud profile is obtained, we perform a PSF-weighted average of the cloud properties on the BBR footprint, obtaining the "true" cloud information for the three test frames. These new cloud inputs (M-CM) are ingested in the BMA-FLX processor to assess the model performance against the RTC-based radiative flux derived for the test frames. The simulated fluxes are

obtained from the RTC employed to produce the L1 EarthCARE radiance data from the GEM results as part of the test frames (Donovan et al., 2023). Figure 3 presents the cloud properties in the test frames derived from model truth fields at the MSI grid. Figure 4 shows the surface types as they come from the RT model. The fresh snow and sea ice surfaces have been used to replace the values provided in the operational X-MET product.

Table 3 presents the results of the radiative flux retrieved by the BMA-FLX processor using model truth inputs compared to

the model truth flux for two BBR spatial resolutions available in the product. The table includes the RMSE, standard deviation (stddev), and Mean Bias Error (MBE) for the solar and thermal fluxes obtained in the three test frames. Results are further divided into fore, nadir, aft BBR viewing angles, and the combined flux. Across all scenes, BMA-FLX SW flux results for the aft view and the combined approach generally show better agreement with the model truth flux, as indicated by lower RMSE and stddev. The combined approach results in better error metrics than those for the individual views. This is particularly

relevant for the Hawaii scene with the highest optically thick clouds, where the combined approach results in significantly better error metrics than those of the individual fluxes, indicating better overall agreement.

As expected, given the lower anisotropy of the radiance field for the off-nadir views (Suttles et al., 1989), the aft and fore views present the lowest differences with respect to the model truth in the LW fluxes. The combined approach tends to have even lower RMSE and stddev, indicating superior overall performance. Biases are consistently negative across all scenes, but

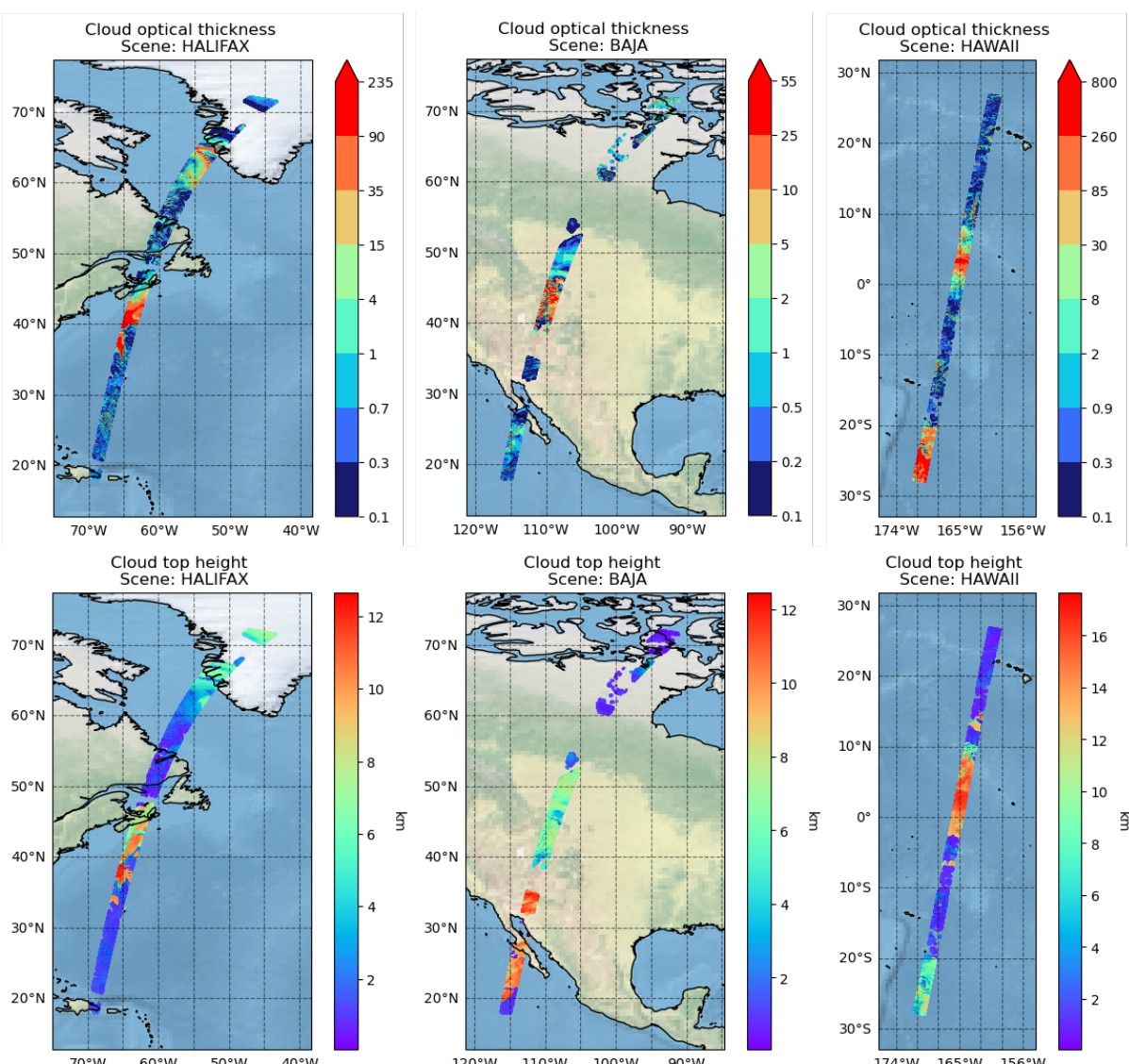

**Figure 3.** MSI-like model truth cloud optical thickness and cloud top height for the three scenes. Colorbars ending with an arrow are logarithmic and have cloud optical thickness values greater than the upper tick label plotted in red. The cloud properties have been calculated using a COT threshold of 0.1.

are relatively small. The combined view consistently provides the best overall agreement with the model truth flux for both solar and thermal fluxes across different scenes.

Figure 5 showcases a comparison between combined SW and LW radiative fluxes derived from the BMA-FLX processor, which has ingested model truth cloud fields, and the RT model truth flux for the BBR standard resolution. The analysis is performed for the different scenes. The time series plots (top row for SW and third row for LW) show the model truth

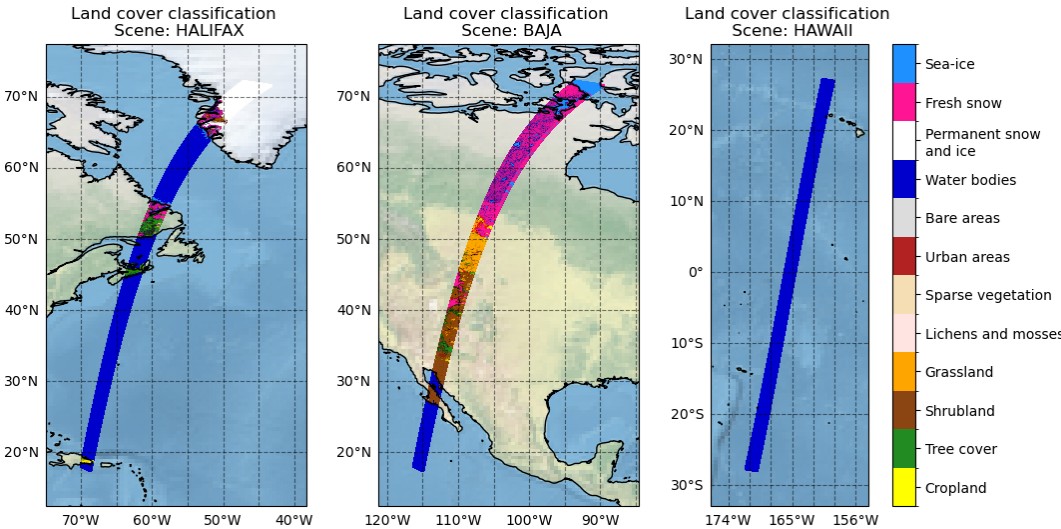

**Figure 4.** Land use land cover (LULC) classification from the ESA Climate Change Initiative (CCI) Land Cover project. Land cover codes are defined in the Land Cover CCI Product User Guide v2 (ESA, 2017). The LULC classification has been simplified to enhance the presentation of the results. In addition, two new categories are presented, them being fresh snow and sea ice.

flux (black line) and the BMA-FLX derived flux (red dots) for each scene. The cloud cover from the model truth (ranging from clear to overcast),the surface classification from GLCC and new X-MET, all employed in the ADM scene identification (e.g., water bodies, forests, savannahs, etc.), are presented at the bottom of the plots. GLCC database provides a surface classification of seven permanent classes. Two dynamic classes, fresh snow and sea ice, are derived from the snow depth and sea ice cover parameters of the X-MET data and added to the classification. The inclusion of these classes enhances

the understanding of how these factors might influence flux retrievals. These comparisons are complemented by statistical measures (mean, standard deviation, and RMSE) to quantify the agreement between the model and the truth data. Additionally, the plots highlight discrepancies exceeding the $10 \ \mathrm{Wm}^{-2}$ threshold defined by the EarthCARE radiative closure experiment. This comparison is shown as a "pre-launch" numerical assessment experiment for validating the performance and reliability of the BMA-FLX processor in diverse environmental conditions. Figure 5 demonstrates a generally good agreement between the

retrieved fluxes and the model truth fluxes. Deep convective clouds, along with optically thick ice and water clouds, are well-represented in both LW and SW retrieved fluxes. However, the greatest discrepancies are observed in cloud-free and partially covered regions. Broken clouds pose significant challenges for both SW and LW retrieval algorithms in real-world scenarios due to the complex interaction of reflected and emitted radiation. SW fluxes retrieved in regions with simulated broken clouds tend to be noisy and less reliable. It is important to note that these fluxes, derived from ADMs based on satellite measurements,

are unlikely to adhere to the radiance-to-flux relationships seen in the plane-parallel simulations used as the reference in this study. LW flux retrievals appear to have a high frequency variations not observed in the simulations that are flattened out when increasing the averaging region. Thermal fluxes obtained for the assessment domain resolution (5 JSG x 21 JSG pixels) smooth

| Assessment Domain resolution | | | | | | | | | | | | |
|---|---|---|---|---|---|---|---|---|---|---|---|---|
| Scene | Halifax | | | | Baja | | | | Hawaii | | | |
| SW | fore | nadir | aft | comb | fore | nadir | aft | comb | fore | nadir | aft | comb |
| RMSE ($\mathrm{Wm}^{-2}$) | 22.8 | 11.2 | 6.9 | 7.1 | 19.6 | 22.8 | 12.9 | 12.7 | 18.7 | 16.3 | 17.4 | 12.3 |
| stddev ($\mathrm{Wm}^{-2}$) | 19.1 | 10.2 | 6.5 | 6.8 | 19.1 | 22.4 | 10.4 | 12.2 | 16.6 | 15.3 | 15.1 | 12.0 |
| MBE ($\mathrm{Wm}^{-2}$) | 12.4 | -4.8 | -2.2 | -2.1 | -4.4 | 4.4 | -7.6 | -3.6 | -8.7 | -5.8 | 8.6 | -2.6 |
| LW | fore | nadir | aft | comb | fore | nadir | aft | comb | fore | nadir | aft | comb |
| RMSE ($\mathrm{Wm}^{-2}$) | 2.6 | 4.1 | 2.1 | 2.2 | 4.1 | 6.2 | 3.7 | 3.2 | 4.1 | 5.6 | 4.1 | 4.1 |
| stddev ($\mathrm{Wm}^{-2}$) | 2.4 | 3.3 | 2.0 | 1.8 | 3.9 | 6.1 | 3.6 | 3.0 | 4.0 | 5.2 | 3.9 | 3.7 |
| MBE ($\mathrm{Wm}^{-2}$) | -0.8 | -2.4 | -0.7 | -1.3 | -1.1 | -1.1 | -1.0 | -1.1 | -1.0 | -2.2 | -1.2 | -1.6 |
| Standard resolution | | | | | | | | | | | | |
| Scene | Halifax | | | | Baja | | | | Hawaii | | | |
| SW | fore | nadir | aft | comb | fore | nadir | aft | comb | fore | nadir | aft | comb |
| RMSE ($\mathrm{Wm}^{-2}$) | 22.7 | 11.0 | 6.4 | 6.8 | 17.9 | 19.9 | 11.5 | 11.2 | 16.9 | 14.8 | 15.2 | 9.3 |
| stddev ($\mathrm{Wm}^{-2}$) | 19.4 | 9.8 | 6.0 | 6.4 | 17.4 | 19.6 | 8.7 | 10.6 | 14.2 | 13.4 | 12.9 | 8.8 |
| MBE ($\mathrm{Wm}^{-2}$) | 11.9 | -4.9 | -2.3 | -2.3 | -4.4 | 3.7 | -7.6 | -3.6 | -9.2 | -6.2 | 8.1 | -3.1 |
| LW | fore | nadir | aft | comb | fore | nadir | aft | comb | fore | nadir | aft | comb |
| RMSE ($\mathrm{Wm}^{-2}$) | 2.9 | 4.1 | 2.1 | 2.3 | 4.6 | 6.7 | 4.2 | 3.9 | 5.6 | 7.0 | 5.5 | 5.6 |
| stddev ($\mathrm{Wm}^{-2}$) | 2.8 | 3.4 | 2.0 | 1.9 | 4.4 | 6.6 | 4.1 | 3.7 | 5.5 | 6.6 | 5.4 | 5.3 |
| MBE ($\mathrm{Wm}^{-2}$) | -0.7 | -2.4 | -0.7 | -1.3 | -1.1 | -1.2 | -1.1 | -1.2 | -1.0 | -2.2 | -1.2 | -1.6 |

**Table 3.** Statistics of the BBR flux estimation compared to the model truth flux using the model truth cloud fields as inputs for the BMA-FLX processor in the three test frames (Halifax, Baja, Hawaii) and each of the BBR views (fore, nadir, aft) and the combined approach for the Assessment Resolution (5 JSG x 21 JSG pixels) and the Standard Resolution (10km x 10km). A COT threshold of 0.1 has been used to derive the cloud properties.

the response, which contributes to the success of the radiative closure. Comparisons over clear-sky scenes are constrained by inherent differences between the surface definitions used in the RTC model derived from the ESA Climate Change Initiative
Land Cover and the BMA-FLX surface definition based on GLCC. This issue is particularly pronounced for land surfaces covered by fresh snow and sea ice, which showed the highest discrepancies before replacing the meteorological X-MET values by the model truth snow/ ice conditions.

## 3.3 End-to-end uncertainty assessment

The end-to-end verification of the BMA-FLX processor was evaluated using three test frames created for EarthCARE. In this
analysis, the BMA-FLX processor ingests the MSI and ATLID products retrieved by the L2 EarthCARE processors. A detailed

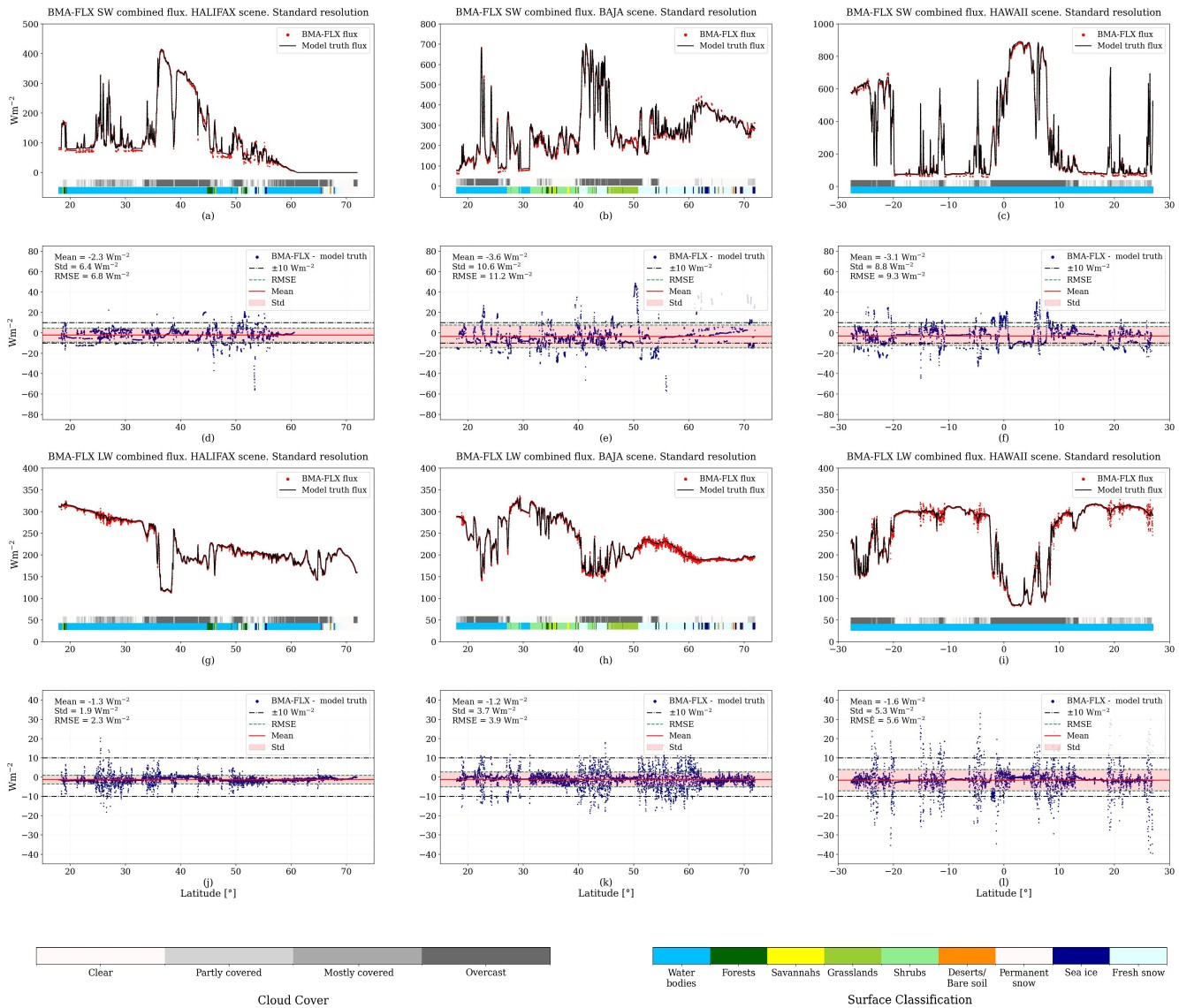

**Figure 5.** Comparison between BMA-FLX flux retrievals incorporating model truth cloud fields and the corresponding model truth fluxes, across the three scenes. The plots illustrate the combined TOA solar and thermal fluxes retrieved by the BMA-FLX processor and their alignment to model truth fluxes. Each time series includes the cloud cover and surface classification information. The cloud properties have been calculated using a COT threshold of 0.1.

description of the L2 products derived from the L1 data (Donovan et al., 2023), produced for these scenes, is available in the respective papers of this special issue.

The results of the comparison between the BMA-FLX and the model truth fluxes are presented in Table 4 for both the Assessment Domain resolution (5 JSG x 21 JSG pixels) and the Standard resolution (10 km x 10 km). The LW results remain consistent with those in the previous section because the LW algorithm utilizes broad and narrow-band radiances independently of L2 cloud retrievals. This independence from MSI retrievals is a significant advantage of the LW algorithm, ensuring that its ADM estimates are not affected by potential errors in other retrieval processes. In SW, the impact of using cloud properties retrieved by M-COP and M-CM is minimal in the Halifax scene. However, in the Hawaii scene, the flux results for the aft and nadir views are significantly higher compared to the analysis using model truth cloud fields (Table 3). This discrepancy arises from the MSI's cloud mask retrieval failing to accurately detect the cloud fraction in the tropical convective system at the center of the Hawaii simulated orbit. Selecting an incorrect SW ADM for very bright cloud scenes significantly impact the average metrics. The combined results for Baja show a decline compared to the previous analysis. The flux results for aft, fore, and nadir views exhibit greater uncertainties compared to those obtained with the true cloud profiles and the better discrimination of fresh snow cover and ice over water. This is likely due to the challenges faced by the M-CLD processor in retrieving cloud information from the simulated cloud fields over snow surfaces above 50° N. Overall, the algorithm for combining the view-based fluxes performs exceptionally well in mitigating the impact of incorrect retrievals from the nadir, aft, and fore models.

Figure 6 shows the combined fluxes retrieved with the BMA-FLX processor alongside the model truth fluxes for the Halifax, Baja, and Hawaii scenes. Unlike Figure 5, the plots in this section present fluxes derived from the L2 EarthCARE operational processors. To facilitate interpretation, both cloud cover and surface classification are included. The results in this figure are significantly worse than those in Figure 5.

As previously discussed, the main discrepancies stem from differences between the model truth cloud fields and the M-CLD retrieved cloud properties for the simulated scenes, which affect the selection of the SW ADM and, consequently, the accuracy of the flux estimation. For instance, in Hawaii, the tropical convective system near the center of the scene (0-5° N) is classified as overcast according to the model truth cloud fields, whereas the M-CM cloud mask reports partly or mostly covered scenes, resulting in flux differences exceeding 80 $\mathrm{Wm}^{-2}$. Another significant discrepancy is observed in Baja, around 50° N, where there is a considerable difference between the model truth and retrieved clouds. Similarly, in Halifax (48° N), the SW retrieval algorithm classifies the region as partly clear based on the MSI retrievals, while the model truth indicates clear conditions. The ADM selected for clear-sky reports SW flux values much closer to the model truth than the MSI-based retrieval. Under clear conditions, the main discrepancies arise from differing assumptions about fresh snow and sea ice surfaces. In certain regions, the simulations indicate the presence of fresh snow cover or ice over water, which is not captured in the X-MET data. .

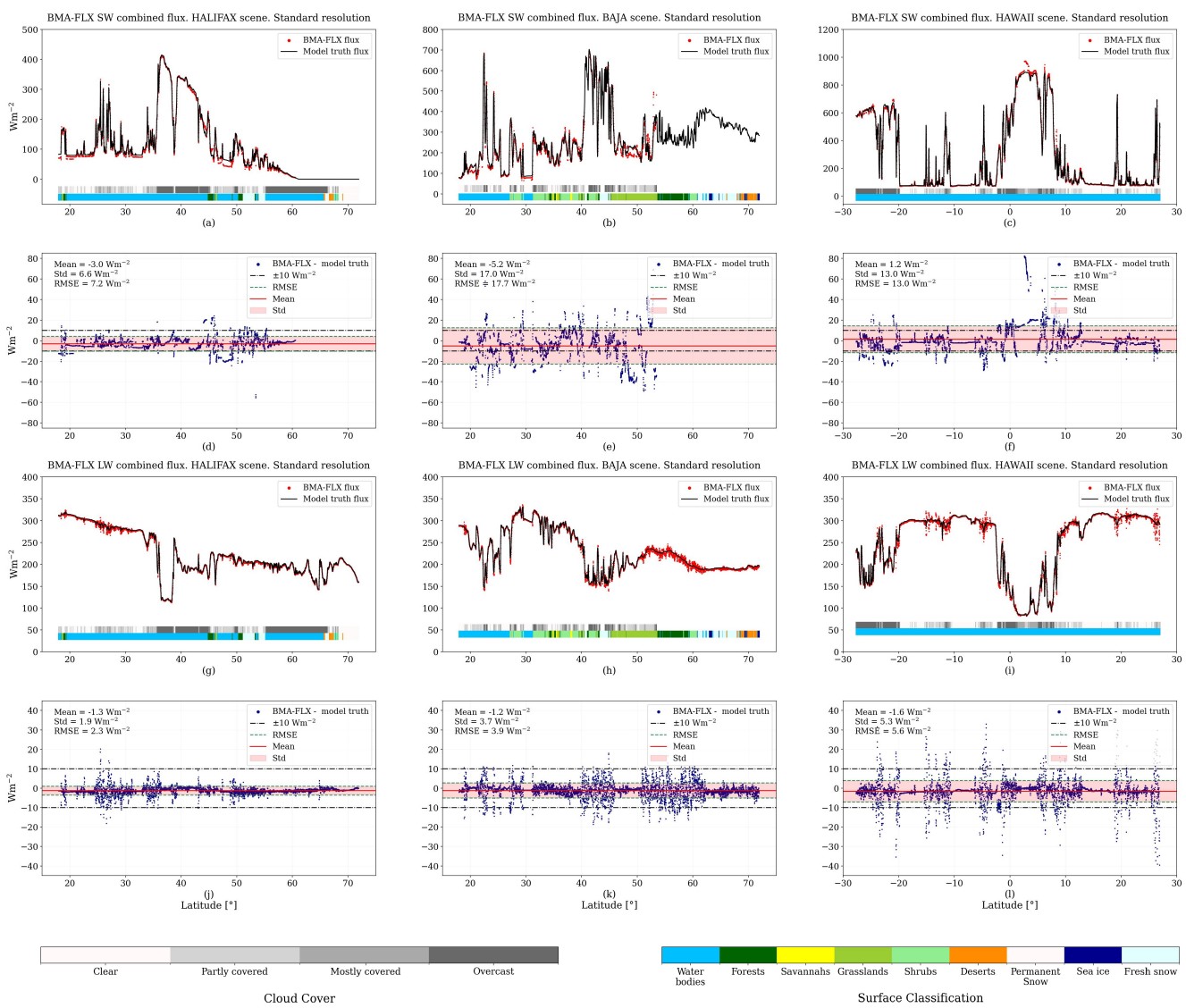

**Figure 6.** Comparison between BMA-FLX flux retrievals ingesting EarthCARE products and the corresponding model truth fluxes, across the three scenes. The plots illustrate the combined TOA solar and thermal fluxes retrieved by the BMA-FLX processor and their alignment to model truth fluxes. Each time series includes the cloud cover and surface classification information.

| Assessment Domain Resolution | | | | | | | | | | | | |
|---|---|---|---|---|---|---|---|---|---|---|---|---|
| Scene | Halifax | | | | Baja | | | | Hawaii | | | |
| SW | fore | nadir | aft | comb | fore | nadir | aft | comb | fore | nadir | aft | comb |
| RMSE (Wm$^{-2}$) | 23.4 | 12.2 | 7.3 | 7.9 | 26.3 | 26.7 | 17.6 | 17.5 | 17.9 | 46.8 | 63.4 | 15.4 |
| stddev (Wm$^{-2}$) | 19.1 | 11.0 | 6.8 | 7.3 | 26.2 | 26.6 | 15.1 | 16.6 | 17.6 | 43.9 | 62.1 | 15.3 |
| MBE (Wm$^{-2}$) | 13.4 | -5.2 | -2.7 | -2.9 | -2.0 | 1.7 | -9.1 | -5.5 | -3.0 | -16.0 | 12.6 | 1.7 |
| LW | fore | nadir | aft | comb | fore | nadir | aft | comb | fore | nadir | aft | comb |
| RMSE (Wm$^{-2}$) | 2.6 | 4.1 | 2.1 | 2.2 | 4.1 | 6.2 | 3.7 | 3.2 | 4.1 | 5.6 | 4.1 | 4.1 |
| stddev (Wm$^{-2}$) | 2.4 | 3.3 | 2.0 | 1.8 | 3.9 | 6.1 | 3.6 | 3.0 | 4.0 | 5.2 | 4.0 | 3.7 |
| MBE (Wm$^{-2}$) | -0.8 | -2.4 | -0.7 | -1.3 | -1.1 | -1.1 | -1.0 | -1.1 | -1.0 | -2.2 | -1.2 | -1.6 |
| Standard Resolution | | | | | | | | | | | | |
| Scene | Halifax | | | | Baja | | | | Hawaii | | | |
| SW | fore | nadir | aft | comb | fore | nadir | aft | comb | fore | nadir | aft | comb |
| RMSE (Wm$^{-2}$) | 22.9 | 12.0 | 6.7 | 7.2 | 25.6 | 24.1 | 16.5 | 17.7 | 16.1 | 47.0 | 64.7 | 13.0 |
| stddev (Wm$^{-2}$) | 19.0 | 10.7 | 6.0 | 6.6 | 25.5 | 24.1 | 14.0 | 17.0 | 15.7 | 44.0 | 63.3 | 13.0 |
| MBE (Wm$^{-2}$) | 12.8 | -5.5 | -2.8 | -3.0 | -1.1 | 0.9 | -8.7 | -5.2 | -3.4 | -16.6 | 13.3 | 1.2 |
| LW | fore | nadir | aft | comb | fore | nadir | aft | comb | fore | nadir | aft | comb |
| RMSE (Wm$^{-2}$) | 2.9 | 4.1 | 2.1 | 2.3 | 4.6 | 6.7 | 4.2 | 3.9 | 5.6 | 7.0 | 5.5 | 5.6 |
| stddev (Wm$^{-2}$) | 2.8 | 3.4 | 2.0 | 1.9 | 4.4 | 6.6 | 4.1 | 3.7 | 5.5 | 6.6 | 5.4 | 5.3 |
| MBE (Wm$^{-2}$) | -0.7 | -2.4 | -0.7 | -1.3 | -1.1 | -1.2 | -1.1 | -1.2 | -1.0 | -2.2 | -1.2 | -1.6 |

**Table 4.** Statistics of the BBR flux estimation errors using the outputs of the processor chain for the three test frames (Halifax, Baja, Hawaii) and each of the three views of the BBR (fore, nadir, aft) for the SW and LW fluxes and the combined flux for the Assessment Resolution (5 JSG x 21 JSG pixels) and the Standard Resolution (10km x 10km).

## 4   Conclusions

This paper describes the algorithm used by the BMA-FLX processor to estimate top-of-atmosphere solar and thermal fluxes using radiance measurements from the BBR and MSI, along with cloud properties from L2 retrievals. Fluxes are independently estimated for each of the three BBR views, and a combined flux is provided by integrating all three estimates in a common atmospheric layer. For SW flux, an artificial neural network model is trained with ADM-based fluxes from the CERES instrument and MODIS radiances and retrievals. LW flux estimates use ADMs constructed from MSI simulated measurements, employing multiple second-order polynomial regressions.

An end-to-end evaluation of the processor was conducted using three synthetic EarthCARE frames, which included L1 data from instrument models and L2 data from EarthCARE retrieval models, along with corresponding RTC-derived fluxes. The

assessment also included an evaluation of the processor's uncertainty, using cloud inputs derived directly from the radiative transfer computations.

Using cloud and snow/ ice properties from EarthCARE processors introduces additional uncertainties. Despite this, the combined flux estimates demonstrated stable results. While the retrieved fluxes from each of the BBR views exhibited significant differences when compared to the model truth fluxes, the combined approach significantly reduced these discrepancies. The combined flux consistently provided the best overall agreement with the model truth flux for both solar and thermal fluxes across all scenes. This approach is particularly advantageous in reducing biases and improving accuracy, especially under challenging conditions, such as the super tropical convective systems in the Hawaii test scene.

In the end-to-end uncertainty assessment of the standard resolution product, the LW fluxes demonstrated strong alignment with the model truth fluxes across all three scenes, benefiting from reduced anisotropy in the oblique BBR views. The RMSEs for these scenes were consistently below $6\,\mathrm{Wm^{-2}}$. In contrast, the SW fluxes showed greater deviations from the model truth, primarily due to the more complex anisotropy of solar radiation footprints and their dependence on cloud-retrieved fields. The RMSEs for the combined fluxes varied from $7\,\mathrm{Wm^{-2}}$ in the Halifax scene to $18\,\mathrm{Wm^{-2}}$ in the Baja scene.

Despite these challenges, the BMA-FLX product represents a significant advancement towards achieving the mission's goals and provides considerable scientific value. The mission's objectives extend beyond meeting specific RMSE thresholds to encompass broader scientific aims, such as understanding radiative processes. Even with higher uncertainties in certain scenarios, the data collected contribute valuable insights into the Earth's radiation budget.

Instances where the error metrics exceeded the $10\,\mathrm{Wm^{-2}}$ threshold suggest that achieving the radiative closure goal will be challenging, requiring both improvements to the cloud property retrieval and the BMA-FLX algorithm, along with improvements to comprehensive data generated by the mission. All these changes will enhance our understanding of how to meet the mission's objectives and offer potential for further refinement and optimization of the BMA-FLX product.

It is important to note that these results are based on a simulation environment, with likely inaccurate simulated fluxes used for 'truth', particularly for broken clouds conditions. Further validation of the BMA-FLX products will occur during the Commissioning Phase. This phase will involve evaluating input ingestion with actual retrievals from X-MET, BBR, MSI, and ATLID data, and will include thorough testing of flux retrieval accuracy, the parallax algorithm, and the reference level methodology.

For future improvements in the ADMs, we plan to test the use of AOD and albedo climatologies in clear-sky conditions. The inclusion of AOD climatology might introduce significant uncertainty due to the large spatio-temporal variability of aerosols, potentially causing discrepancies between the actual AOD for any given BBR measurement and the climatology, thus affecting anisotropy. We will evaluate the operational use of the EarthCARE MSI's AOD product (M-AOT, Docter et al. (2023)). Additionally, the SWIR imager channels (1.65 and 2.21 $\mu$m), crucial for determining cloud parameters, are not currently utilized in the SW ADM. Future updates will include these bands to enhance cloud field characterization. Furthermore, alternative algorithms that leverage the multi-angular capabilities of the BBR will be explored in future iterations of the BMA-FLX processor to complement the current method of integrating fluxes from each BBR telescope.

In conclusion, the BMA-FLX processor demonstrates a significant advancement in flux estimation. By combining flux estimates from multiple BBR views, it effectively reduces uncertainties present in individual retrievals. This combined flux approach offers a robust and reliable method for calculating fluxes across various atmospheric conditions. The BMA-FLX processor's ability to handle the complexities of the anistropic radiance measurements provides accurate top-of-atmosphere flux estimations for both SW and LW radiation. This leads to reliable data for the EarthCARE's radiative closure and and

brings the mission closer to achieving its scientific and operational goals..

*Data availability.* The EarthCARE L2 demonstration products from the simulated scenes, including the BMA-FLX product discussed in this paper are available from https://doi.org/10.5281/zenodo.7728948 (van Zadelhoff et al., 2023a)

*Author contributions.* A.V.B. and C.D. conceived the idea for this manuscript. A.V.B. and N.C designed the LW flux retrieval algorithm. A.V.B. N.C and E.B. developed the LW algorithm and the BMA-FLX LW processor. A.V.B. validated the LW algorithm. C.D. developed
the theory of the SW algorithm, created and validated the SW algorithm and coded the BMA-FLX SW processor. E.B is responsible for the pre-processing and PSF weighting for the SW and LW processors. C.D. supervised the projects funding this activity. N.M. helped in the writing and proofreading of the manuscript. C.S.M. prepared graphical material of the manuscript, and contributed to the work done for obtaining the model truth fields. All authors discussed the results and contributed to the final manuscript.

*Competing interests.* The authors declare that they have no conflict of interest.

*Acknowledgements.* We would like to express our gratitude to Dr. Tobias Wehr, who passed away on February 1, 2023, for his support and invaluable guidance throughout our many years of work together. We are also indebted to our former colleagues Raquel García Marañón, Alessandro Ipe and Florian Tornow for their contributions in the development of the BMA-FLX processor. The CERES SSF data used in this study were obtained from the NASA Langley Research Center Atmospheric Science Data Center.

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
