# Peer review of "Retrieval of top-of-atmosphere fluxes from combined EarthCARE LiDAR, imager and broadband radiometer observations: the BMA-FLX product"

_EGUsphere, 2024_

## Referee Comment (RC1)

**Review of "Retrieval of top-of-atmosphere fluxes from combined EarthCARE LiDAR, imager and broadband radiometer observations: the BMA-FLX product" by Velázquez Blázquez et al.**

**25 June 2024**

**General comments**

This paper describes the BMA-FLX product that is used to retrieve radiative fluxes from the broadband radiometer (BBR) on the EarthCARE satellite mission. The product retrieves a SW and LW radiative flux for each of the three view angles of the BBR and then averages them to provide a best estimate of the fluxes. This best estimate is compared to the calculated fluxes for some example scenes, with an overall goal of obtaining radiative closure within 10 W/m2.

It is important to document the details of the BMA-FLX product so that future users of EarthCARE data can understand how the data is generated. The paper is thorough and mostly well written. However, I believe some substantial additional explanations are needed. I am also left concerned as to whether the performance of the product is sufficient to meet the EarthCARE goals based on the analysis presented. I outline these concerns in my specific comments below. After addressing these comments, I believe the study could be appropriate for publication in AMT.

**Specific comments**

Terminology: There is some debate in the ERB community as to whether "flux" is the correct term for the quantity considered in this paper that has units of W/m2. While "flux" is widely used, some argue that it is fundamentally wrong and that "flux density" is correct. Another alternative, "irradiance", is becoming more widely used because it is less ambiguous. I do not insist that the authors rename their product, but I do think that it is worth mentioning in the introduction that several different terms exist in the literature. These terms are often used interchangeably within the scientific community with some debate, but they all refer to the same quantity.

L57: It would be good to add a sentence or two to justify why the EarthCARE radiative closure goal is 10 W/m2, otherwise this seems rather arbitrary. What is the physical origin of this requirement? What does 10 W/m2 achieve that eg. 15 W/m2 does not? What limits a more ambitious goal of eg. 5 W/m2? Also, is this requirement defined based on the RMSE, which seems to be a focus of the results later in the paper and in the conclusions?

L66-69: Following from the previous comment, a more general question is why is the EarthCARE radiative closure goal defined in terms of radiative flux? As noted in the paper, the conversion from radiance to flux inevitably introduces a substantial uncertainty. Unless I am missing something, it seems entirely possible to use a radiative transfer model to calculate *radiances* at the three view angles of the BBR using the scene properties derived from the other EarthCARE instruments, and compare those radiances to the directly observed radiances from the BBR. While flux is usually the quantity of most interest for atmospheric energetics, I think for the purpose of radiative closure, performing the closure in radiance-space provides a much tighter constraint and avoids the introduction of ADM uncertainty. Some mention of the justification to do radiative closure with fluxes in the introduction of this paper would be helpful to further motivate the BMA-FLX product.

L77-81: It is understandable that the algorithm needs to meet contingency requirements of the mission, but at the same time it does seem like a missed opportunity that the flux retrieval does not take advantage of the multi-angle views of BBR directly to constrain the radiance to flux conversion. Could this be mentioned as a potential future research activity?

L81: I think there is an error here. The outgoing flux should NOT depend on the viewing geometry. Do the authors mean the solar geometry?

L103-105: These two sentences do not make sense to me. I think I know what the authors mean, but I suggest revising the wording.

L108: I do not understand why the scene type is defined separately for forward and backward scattering directions. Please clarify in the paper why this is needed.

L134-135: Using an AOD climatology might be an important source of uncertainty here. This is because the large spatio-temporal variability of aerosol means that the actual AOD for any given CERES measurement can be quite different from the climatology and therefore create a different anisotropy. Do the authors think this is important and have they considered using an aerosol retrieval product from MODIS to match instantaneously with the CERES measurements?

L143: I like the idea of using the MSI radiances directly as an input to the ANN because they contain information about cloud properties without introducing uncertainty from an intermediate cloud property retrieval. That said, I am a bit surprised by the choice of bands: 0.67, 0.865, 10.8, and 12.0 μm. The most relevant cloud properties for anisotropy are cloud fraction, optical depth, phase, and effective radius. Are the authors confident that information about these cloud properties is sufficiently represented by the 4 MSI bands that are chosen? Some supporting references would help.

L153-159: If I understand correctly, different datasets are used to determine the surface type for the scene identification (GLCC and X-MET) than what is used in the CERES training data (IGBP and NSIDC). Why not use the same datasets to minimize errors associated with different dataset definitions?

L232: Some more information about the ADM uncertainty is needed. Where does this uncertainty estimate come from?

Fig. 3: The cloud optical depth would be easier to visualise with a much more limited colour bar, or maybe a non-linear colour bar. Otherwise, the vast majority of the data points are in a very limited range and it is difficult to see the variability.

Figure 4: It is not very useful to plot the land cover codes and refer the reader elsewhere for the definition of the codes. Can the codes be replaced by the definitions directly in the plot? Or put in a nearby table?

Figure 4 and 5: They are referenced in reverse order in the text. I suggest switching the order.

392-393: Are the significantly worse results in Figure 6 concerning? I believe this is the most relevant uncertainty estimate because when the algorithm is applied to actual EarthCARE observations, BMA-FLX will need to use the M-CLD retrieved cloud properties, correct?

L423, Table 1, and Table 2: There are several instances for both the assessment domain and the standard resolution where the combined RMSE in the SW exceeds 10 W/m2. If the RMSE is indeed the

relevant quantity (see earlier question), I interpret this to mean that the radiative closure goal will regularly be exceeded, which then leads me to question whether BMA-FLX is sufficient to achieve the closure goal of the mission. I expect that this is not the impression that the authors would like to leave the reader. I think the conclusions need some additional discussion to relate these results back to the mission requirements and what this means for the adequacy of the BMA-FLX product.

---

## Referee Comment (RC2)

**Review of: Retrieval of top-of-atmosphere fluxes from combined EarthCARE LiDAR, imager and broadband radiometer observations: the BMA-FLX product**

The paper presents a description of the BM-FLX SW and LW radiance to flux conversions including the determination of the required cloud information from the MSI cloud properties. Using EarthCARE test frames it goes on to evaluate the error component in the fluxes due to the conversion process for both the ideal case of perfect cloud information and for the case of cloud information derived from other EarthCARE products. The work presented provides important information to users regarding the basis of the radiance to flux conversion and the evaluation undertaken is a valuable and necessary exercise providing an initial look at the flux performance compared to modelled fluxes.

The paper is generally well structured but in terms of clarity I think that some additions and modifications are needed in the description of the method in section 2. Some context to the evaluation would also help elucidate the scope and limitations of the comparisons presented. The goal of being able to estimate flux to 10 Wm$^{-2}$ at the 10km by 10km for both the SW and LW is a very challenging requirement (particularly in the SW). Without knowing the details of the specification it is not immediately clear to me if this is k=1, 2 or 3 requirement on the error or an upper limit so I don't know exactly how to relate it to your results. If possible it would be good to clarify this, although I realise that these things are not always as clearly specified as they might be and may require a bit of interpretation on your part, in which case I would suggest you explain your interpretation to provide some indication of the bar you are trying to meet. I note that you mention the requirement in relation to achieving radiative closure to within 10 Wm-2 (which I appreciate is the origin in the requirement within the EarthCARE mission) but this is possibly even harder to relate to your results and to be honest I think not clearly related even at mission specification level when deriving the BBR accuracy requirement, so it may be easier to stick to discussing the flux product requirements, however this is up to you. Whatever the specifics I think it would be useful for users to be able to understand how these results indicate they should employ the data for their chosen purpose and how they might identify fluxes that meet a certain quality criteria or a means by which they may be able to meet a specific criteria. To this end it would be useful to know if quality indicators in the fluxes themselves can provide some information on accuracy of the fluxes allowing users to select an appropriate subset. Alternatively or additionally it would be useful to know if averaging over a larger domain provide generally better results (this is implicit in the bias and seems likely looking at the results and I think knowing at what scale you may meet a specific criteria would be useful). Also some discussion of how representative the errors are likely to be (given you are comparing to model fluxes), and discussion of the way forward regarding the effect of the scene identification errors would be helpful. For example, are the scene errors because these products don't meet their specifications and will be improved? Is it because the model radiances aren't as realistic as the retrieval and thus the errors are not representative of the real world or are these errors that you have to live with and can you adjust / tune the scene id to cope with them?. I don't think that addressing these things constitutes a major change to the paper in any sense.

I detail these issues below along with noting some possible issues with figures 3, 5 and 6 which may be incorrectly displayed.

**Main points:**

Section 2.

Lines 59 to 84. At this point in the paper you are discussing the definition of flux generally (both SW and LW), however the formulation you present is specific to the SW so is a bit confusing. I think this section should be general to both SW and LW or needs to become part of the SW section below and then be repeated for the LW. I would suggest you can generalize by stating that you are integrating the radiance field over viewing zenith and azimuth (this is true for the both LW and SW). The radiance depends on the scene the viewing geometry and additionally for the SW the solar geometry. Note in this case you would not use relative azimuth but viewing azimuth as the integration term. You would in this case also need to avoid anything specific to the shortwave method. Alternatively, you will need to formulate a separate discussion and equation for LW and SW but I don't think this is needed.

Lines 67 you 69. This is phrased a bit strangely and I am really not sure about the points these two sentences are making. Converting radiance to flux using a model of the radiation anisotropy is the method you are employing. As opposed to calculating flux from properties retrieved from radiances for example. The model used to represent non uniform variation of the radiation field (i.e. its anisotropy), is an angular distribution models (ADMs) in a general sense. You state 'The ADMs have accurately represented this variation (Su et al., 2016) and so are used as the basis for flux retrieval'. My understanding is that you are not actually directly using **the** particular ADMs described in Su et al. although they may be relevant to the accuracy of the fluxes you are using this does not automatically imply any accuracy of the ADMs you derive from those fluxes if you follow a different classification system for example. If you just mean by this sentence that ADMs can be accurate, this may be true but that some angular distribution model has been used successfully before doesn't have any bearing on the accuracy of any other ADM. So I'm a bit lost and think maybe a few things are not quite said as intended, particularly bearing in mind that at this point you are talking about both the SW and LW flux derivation. I think you want to make the following points but I'm not sure if you really want to make them all here:

- You are developing your own ADMs to represent scene anisotropy and enable fluxes to be derived from flux.
- For the SW you are using CERES fluxes to develop your model, these employ their own distinct ADMs which are of established accuracy (although this is only relevant for the SW I think so that will need to be clarified)

Line 70 to 72. I think this needs to be rephrased. An ADM doesn't estimate the flux, it represents [an estimate of] the relative angular distribution of the radiation field and can be used for the derivation of flux from a single radiance measurement. Similarly an anisotropic factor at a given set of angles does not define the ADM but can be derived from it to enable the conversion of a radiance at a given angle to flux.

Section 2.1.3 I think it is a difficult task to try and convey your selection of inputs here via a narrative style I would suggest that a table may be an easier way to convey much of this information listing all the possible inputs the source of the information and then the scene classes where they were included. Similarly, some indication in this or a separate table of the best/chosen inputs, at present I don't think this section makes it clear what the inputs end up being. I think this would clarify a lot of what is trying to be explained and could replace much of the text in this section.

Line 121. 'a predefined list of parameters that influence scene anisotropy is very vague. I assume this relates to the climatology and meteorological data you speak of above. I think a

table listing all the variables considered and some discussion on those eventually considered significant would solve the ambiguity.

Line 122 'BBR-received scattering direction' I'm not sure if you mean each BBR view or if you mean the forward and backwards scattering directions generally (in which case this is part of the scene) or if you mean every bin of the ADM. Can you please clarify.

Line 125 you speak of two methods which 'largely agree' how are they combined and how is any disagreement treated, do you just AND the two sets?

Line 131 where does LAI come from is this a dynamic MODIS value integrated at footprint level or a generic value based on the static classification of the surface. Where do you get this information in the evaluation section it doesn't seem to be discussed.

Line 132 how do you 'consider' the hot spot effect as input, what is used as the input?

Line 133 to 134 you say AOD and wind speed is 'selected as parameters for network training' what do you mean by selected, is this selection the result of testing (i.e. selected by some criteria in which case please explain) or just considered to be important ('chosen')

Line 139 how are MSI-like MODIS radiances created? Is this just the nearest wavelengths or are some spectral adjustments done, are they something you create or a defined product, can you please include further detail or a reference.

Line 140 it's a very plausible assumption but would benefit from an extra line of justification maybe stating which narrow bands you use and pointing out they are the ones used to retrieve cloud properties and thus implicitly contain that information albeit extracted via some other model.

Line 143 could you clarify if the process determines if these are the best or if you choose these.

Line 146 to line 147 how is the 20% of the dataset used for validation chosen and is it truly independent? E.g. every 5th ceres footprint would not be independent every 5th orbit *might* be, but I would suggest that 1 year out of 5 would most likely be the best test. Is the CERES validation dataset the same as the cross check validation or different? How much data is this and how is it chosen. What is the result of this validation what expected best case errors are you expected from the retrieved flux on the basis of this validation?

Line 155 You say the retrieval algorithm uses two surface types and combines them, do you mean you make two separate retrievals for each class individually which are then combined or do you mean there is some sort of combined class possible in the retrieval?

Line 162 what DEM is used, and do you also do this for elevations below 0 (assuming zero is the average Geoide or is it something else? Although I'm a bit confused because surely the coregistration at the surface performed by BBR considers some sort of DEM already.

Line 164 to 166. This sentence is unclear at this point, it becomes clearer after figure 1 is discussed maybe reference this later discussion or consider rearranging such that the 2.1.5 comes before 2.14? Also I would replace cloud properties with cloud fraction as I think this is the only 'property' you are using for scene? If you are also using other properties for input it might be worth clarifying somewhere what is used for that for the oblique view although probably this shouldn't go in the scene section, although I'm not sure it would make sense to

use nadir properties in that case unless that somehow corresponds to what was used in the training.

Line 166 to 175. I am confused by the discussion on cloud top height in this section on scene identification as whilst it has been described as an input it has not been discussed as part of the scene classification. Is this part of the scene classification (in which case it should have been discussed in scene definition section), does it just relate to matching scenes (in which case can you clarify this) or is this really a discussion on inputs in which case it should go elsewhere or the scope of this section should be extended.

Line 183 'providing there is no significant elevation' do the BBR products coregister at the 'surface' or at some average sea level, surely the former which would include elevation of surface?

Line 195 to 196 'CTH...is a reliable estimator for co-registering...in the thermal' was this shown in the AATSR study or some other way how is this know. One might have thought that thin cloud may pose a similar issue in the thermal as in the visible, or that wet atmospheres might may the co-registration point above a low cloud, or in clear sky make the surface a poor point of co-registration.

Line 202 how are the errors in the ADM treated when minimizing the height of co-registration, what happens in cases that are poorly behaved and a minimum isn't found?

Lines 223 to 225. I find this sentence difficult to understand. I think maybe don't use the word 'coordinates' unless you really mean 2d or 3d coordinates and or specify what coordinates you are talking about. So if you just taking about the central pixel in the nadir view, say this, if you are talking about the 3d location of the nadir reference level then specify this, if you mean the location of the ctp in the nadir then again specify.

Line 228 equation 5. Do you calculate this for each combination, i.e. nadir vs fore, nadir vs aft and aft vs fore?

Line 229 this equation does not necessarily limit the range of 0 to 200%, if you limit it artificially maybe this should be specified in the equation by an alternative formulation,i.e. as stated if < 200% and 200% if greater than that.

Line 244 do you have any evidence that using a set of regressions solve the problem and if so why/how. Maybe you need to just state the cause of the issue was a deviation from the fit used for this scene type (rather than an issue for a plane parallel assumption for example which would not be solved but a different regression). Also is this set just the splitting into 5 degress in VZA and 20 Wm-2sr-1 bins or is it something more based on cloud information could you explain somewhere in this paragraph, either state on line 244 that it is radiance and angle separated or explain it is also cloud property dependent and include this addition on line 246.

Line 265 how are uncertainty estimates associated with the fluxes derived this was derived, are they related to the MSI bt uncertainties propagate through the regression used to derive the anisotropy? Do they take into account the scatter about the original regression?

Line 267 can you state what the reference level is in clear sky, is it the surface or somewhere above and does it vary with atmospheric moisture?

Line 280 how it is evaluated from the CERES data? I assume you mean that the ceres data indicate that equal weight should be given to the forward and nadir views in combination. It is

not clear what you are doing here but I suggestion (see the following point) that it might be worth a bit of explanation.

Line 281 I think this requires a bit more comment. You are saying the studies with CERES implies that the LW radiance distribution doesn't behave in a plane parallel way. However you are using plane parallel assumptions to base your radiance to flux conversions on so this would seem potentially concerning. If your comparison with CERES is based on applying your theoretical plane parallel ADMS to forward and nadir and comparing the induvial vs combined result with the CERES empirical ADM conversion then this would support this equal weighting to more closely match the empirical CERES result. But I think this needs explaining and possibly the actual improvement and discrepancies found stated here.

Section 3.

Lines 325 to 329. I think it would be helpful to have a previous section clarifying the two step comparison you will execute in 3.2 and 3.3 and the purpose and scope of the two comparisons and strengths weakness and limitations of these assessments. I think the two step approach you use is reasonable (and in fact wonder why you don't go further and substitute a 'model truth' for the X-MET data. But I think it would be useful to also discuss the limitation and issues of the model truth in the context of your method as whilst it assesses much of the performance it will obviously be limited to how the model fluxes relate to the ADMs which is not necerssarily going to be the same as the real world. You discuss some of this such as surface mis-match when you consider differences but I think it might be helpful to have a broader stage setting consideration of how the model fluxes might not match the SW adms you use, or how assumptions in the longwave such as cloud modelling or surface emissivity might differ between the model the information used to derive your ADMs.

Line 335 'being 0.1 the one with better performance' → '0.1 being found to have the best performance'. I don't understand what is being done here, what determines the best performance? Also why is this not needed to be repeated for the retrieved parameters, this would account for any offset in the retrieval. Will this be redone operationally to determine an operational threshold and if so how will performance be assessed in that case?

Figure 3 – Upper panels please label the colour bar and check that it is correct. The caption states it is cloud optical depth (at 680nm ?) but the scales seem quite extraordinary. I would suggest if you really have COT up to 800 you might consider a logarithm scales as a more useful way to provide information in these figures.

Lines 342 to 353 I think that results in table 1 need to be discussed in terms of the differences between the simulations and the FLX processor assumptions. For example the SW flux retrieval is based on observed radiance distributions, albeit somewhat indirectly (via CERES fluxes based on observationally derived ADMs), in contrast to plane parallel (?) model calculations, can this be an explanation of the variation in the sign of the bias between views in the SW do you think? What might explain the consistent offset in the LW which actually seems worst in the nadir view, is this a consistent difference between the limb darkening effect between the model simulation and your simulations or a surface emissivity effect or something you might expect to persist with real data?

Line 358 Are you saying you use only the GLCC classes and don't use anything equivalent to the X-MET to id fresh snow, later (line 372) you seem to imply you use X-MET would it not be helpful

to put what you use in the ADM here if this is different, maybe you do and you just need to clarify you use GLCC updated where appropriate with X-MET here?

Line 367. I think before you are too hard on your performance under broken cloud conditions you should consider how well these are likely to be simulated. The 'complex interaction' seems to imply a 3d effect but are these considered by the model truth. Or do you mean to say this might be a reference level issue?

Line 370 to 374 It is not immediately obvious which has snow and which doesn't in your assumption vs the model truth, but I think you are saying you don't assume snow because it isn't in the X-MET data but the model truth includes snow. Have you confirmed that if you use snow here that the problem goes away. Is using the X-MET data sensible in this case rather than using the model truth surface?

Figure 5, Although the mean lines look they are probably correct the Std an RMSE plotted on the graphs don't seem to be correctly plotted or at least do not correspond to the values shown plotted around the mean. Furthermore, I think it would be useful to see how the errors relate to any quality indicators on the flux you have (e.g the flux uncertainties from the ADM (equation 6) you have for the SW and or the discrepancy between the fluxes derived from different views or some other measure from the retrieval if it is good or not). Maybe this could be indicated by colour coding the different points that fell above a certain retrieval goodness threshold or by including an additional plot of flux difference against retrieval uncertainty.

Also in the longwave there seems to me to be an indication of noise in the retrieved flux that might be significantly improved if you increased the averaging region (smooth the results seen here). Do you think this is the case. Would it be worth looking at what improvement this wrought to the percentage of points within the 10Wm-2 line, and would be useful information in the context of closure style comparison studies.

Lines 384 to 387. You discuss the differences between the cases but is there indication of why they behave so differently?

Line 387 Is the lack of cloud information from M-CLD just for this test of going to be true operationally and what is the implication if the latter?

Lines 394 to 401 What are the implications of this, is it expected that M-CLD will improve or can/should some adjustment be made to the result before employing them in this manner or are these results a truer representation of the FLX accuracy and indicate its reliance on M-CLD c.f. that used in the classification? Is the 0.1 optical depth threshold found to perform best earlier still valid or does it need to be adjusted?

Line 393 You state the results are significantly worse, they seem to have a lot of scatter to them and I wonder to what extent they might improve if you increased the averaging domain (possibly because the cloud information is more accurate on these scales) or is this a result a biases in the cloud information. Can you relate this to known errors in the cloud information, do they meet their requirements are they expected to improve in orbit or be improved.

Figure 6. As was the case for figure 5 although the mean lines look to correspond to the stated values the lines indicating the RMSE and Std don't seem to be correct. The same points about showing the errors in the context of retrieval uncertainty main in relation to figure 5 apply here, it would be good know if you can tell you are likely to have poorly retrieved fluxes.

Section 4

Can you clarify how your results relate to the goal requirements, I realise this may require some interpretation on your part. I don't think that the std can be related to the 10Wm-2 requirement unless you also consider the effect of the bias. Can points meeting the requirement be identified via a flux quality indicator or by excluding some subset of scenes or by averaging over a larger domain if so what size. Can you add some discussion of what happens next, if the scene id is expected to improve in orbit, if it needs to be altered or if you can tune your flux retrieval.

**Minor corrections:**

Line 3 design → designed

Line 4 remove 'an algorithm' (it is a processor specifically created I assume consisting of several algorithms)

Line 6 to 7 'measurements' It is not clear here if you are talking about the radiances or the fluxes or all the products. It might be clearer to change measurements to radiances if that is the intent, radiance and fluxes of simply all products depending on what you mean.

Lines 7 and 8 'of the atmosphere in cloud condition (reference level)' doesn't make sense. Do you mean '..or in cloudy conditions at a reference level which corresponds to the radiatively most significant vertical layer of the atmosphere'

Lines 12 to 14. 'The radiance to flux conversion algorithms have been successfully validated.....' Make clear this is done here in this paper and state the result obtained in the paper, something along the lines of. 'Validation of the radiance to flux conversion through end-to-end verification using L1 and L2 synthetic data for three EarthCARE orbits. The results .....'

Line 40 (and multiple other occurrences) change 10x10km2 → 10km x 10 km throughout. Could also be 10 by 10 km or square with side of 10km or 100km2 region (it is 100km2 but you are not presenting the equation so I think 10x10km2 is a bit strange. Wehr et al 2023 used 10km x 10km so this might be safest.

Line 44 'are challenging' → is challenging (challenging relates to estimation which is singular)

Line 50 'is created per scene and constructed from'. I'm not sure if you are trying to say that the algorithm is determined from scene stratified observations from CERES and MODIS, or it you are saying that it is a scene dependent algorithm or both, could you please rephrase to clarify.

Line 57 'the mission goal of 10 Wm$^{-2}$' please explain what the goal is.

Lin 62 'integrating the radiance field' need to add somewhere in the sentence that the integration is **over viewing and azimuth angles.**

Line 76 could you state the time period during which the 3 views are obtained to clarify the 'almost simultaneously'. Also I would suggest replacing 'providing a detailed view of scene anisotropy' to 'providing information on scene anisotropy' as whilst three views are helpful, they are not necessarily 'detailed' information.

Line 95, 'CERES instrument' → 'CERES data' (assuming you are using a data product) and state the version of the CERES product used and the years used.

Line 104 'scene definition concludes the number of ANN training sets' I don't think that concludes is the correct word here, do you mean determines?

Line 106 'relies on' → 'consists of'

Line 106 to 107 You say there are 6 static surface types but list 7 in the brackets, so either the number stated or the listing is incorrect

Line 107 'type' → 'types'

Line 108 can you somewhere in this sentence confirm the total number of scene types (I assume this is 72 but it would be good to state

Line 110 Are the NSIDC and NESDIS snow maps combined by you or are they combined by the SSF product? If the latter then change the sentence to say you use the CERES SSF snow infor which is a combination of these, if the former then explain how you combine them.

Line 137 'considered' → 'included'

Line 148 'estimates and original'…'estimates compared to the original…' Do you have this estimate?

Line 152 you mention fore and after BBR observations, but presumably it also classifies the nadir?

Line 189 surely it is 'fore and nadir' from AATSR not 'nadir and aft', the along track view is in the forward direction and obtained before the nadir view.

Line 190 can you include a reference or further detail of the NB to BB used at least the channels employed.

Line 208 equation 4, is there a reason that there is no vza dependence indicated in R and L?

Line 214 'ideally reflected from the same atmospheric domain' what does this mean 'ideally' and 'domain' specifically.

Line 219 what 'internal consistency check' can you please explain what is done, is this just referring to the minimization in 4 or something additional, if additional how is this different from 4?

Line 222 'higher cloud those observed'… not sure if you mean 'higher cloud than those observed' or something else.

Line 235 add to last case 'and $\delta = 0$ for the others'

Line 242 Add reference for the GERB issue with semi-transparent cloud issues: Dewitte et al 2008 https://doi.org/10.1016/j.asr.2007.07.042 and Clearbaux et al 2009 https://doi.org/10.1016/j.rse.2008.08.016.

Line 247 to 250. I think it worth noting here again the absence of a water vapour channel and the that the channel difference fills that role.

Line 252 'allows to obtain' → 'enable sufficiently accurate anisotropy models to be derived from…' Also do we have any idea what the theoretical error is or any reference of what is possible from theory any reference for this assertion even if it is the LW errors achieved in GERB c.f. CERES for clear sky?

Line 253, might be worth stating that azimuthal dependence is negliglbe/neglected before stating R only is a function of VZA.

Line 256 can you put Velazquez et al 2010 on zelando and get a doi to make it accessible?

Line 270 'done at' → 'obtained at a'

Line 271 'would' → 'should' or 'will'

Line 273 'and statistically reducing the' → 'reducing the'

Line 281 'in average' → 'on average'

Line 286 Can you state what cloud properties are used, just fraction and height or something more?

Line 303 'chained...to' → 'interfaces...with'

Line 349 'As expected' is it expected in the model case specifically because it is using the plane parallel assumption can you specify that.

Line 354 Can you clarify here and in the caption to figure 5 and figure 6 if the BMA-FLX results shown are the combined view result.

Line 358 'employed both' → 'both employed'

Line 381 'the longwave results remain consistent...' → 'the longwave results are identical to' (?)

Line 420 'greater dependency' → 'dependency' (given the lw has no dependency.

Line 486 – The doi associated with this paper is for the unpublished version I think it needs to be updated to https://doi.org/10.5194/amt-16-5327-2023 unless you specifically want to reference the unpublished submitted version and not the final publication.

---

## Author Comment (AC2)

**Answers to Referee #1**

The authors thank the referee for the evaluation of the manuscript. Please find our answers (blue text) to the comments (black text) below.

**Review of "Retrieval of top-of-atmosphere fluxes from combined EarthCARE LiDAR, imager and broadband radiometer observations: the BMA-FLX product" by Velázquez Blázquez et al.**

**25 June 2024**

*General comments*

This paper describes the BMA-FLX product that is used to retrieve radiative fluxes from the broadband radiometer (BBR) on the EarthCARE satellite mission. The product retrieves a SW and LW radiative flux for each of the three view angles of the BBR and then averages them to provide a best estimate of the fluxes. This best estimate is compared to the calculated fluxes for some example scenes, with an overall goal of obtaining radiative closure within 10 W/m2.

It is important to document the details of the BMA-FLX product so that future users of EarthCARE data can understand how the data is generated. The paper is thorough and mostly well written. However, I believe some substantial additional explanations are needed. I am also left concerned as to whether the performance of the product is sufficient to meet the EarthCARE goals based on the analysis presented. I outline these concerns in my specific comments below. After addressing these comments, I believe the study could be appropriate for publication in AMT.

*Specific comments*

Terminology: There is some debate in the ERB community as to whether "flux" is the correct term for the quantity considered in this paper that has units of W/m2. While "flux" is widely used, some argue that it is fundamentally wrong and that "flux density" is correct. Another alternative, "irradiance", is becoming more widely used because it is less ambiguous. I do not insist that the authors rename their product, but I do think that it is worth mentioning in the introduction that several different terms exist in the literature. These terms are often used interchangeably within the scientific community with some debate, but they all refer to the same quantity.

Thanks for your comment. This has been mentioned in the introduction as:
Various terms exist in the literature to refer to radiative fluxes, such as flux density and irradiance. Although these terms are often used interchangeably within the scientific community, with some debate, they all refer to the same quantity.

L57: It would be good to add a sentence or two to justify why the EarthCARE radiative closure goal is 10 W/m2, otherwise this seems rather arbitrary. What is the physical origin of this requirement? What does 10 W/m2 achieve that eg. 15 W/m2 does not? What limits a more ambitious goal of eg. 5 W/m2? Also, is this requirement defined based on the RMSE, which seems to be a focus of the results later in the paper and in the conclusions?

Thanks for your comment. Please let us refer to the official and public documentation for EarthCARE to answer your question. The EarthCARE radiative closure's goal of 10 Wm-2 is defined in the EarthCARE Mission Requirements Document (MRD) (Wehr T., 2006). The accuracy requirements placed under the

EarthCARE measurements are there described and justified. We added the corresponding reference to the manuscript in the text for better context.

T. Wehr, editor. 2006. EarthCARE Mission Requirements Document. Earth and Mission Science Division, European Space Agency, https://doi.org/10.5270/esa.earthcare-mrd.2006

L66-69: Following from the previous comment, a more general question is why is the EarthCARE radiative closure goal defined in terms of radiative flux? As noted in the paper, the conversion from radiance to flux inevitably introduces a substantial uncertainty. Unless I am missing something, it seems entirely possible to use a radiative transfer model to calculate *radiances* at the three view angles of the BBR using the scene properties derived from the other EarthCARE instruments, and compare those radiances to the directly observed radiances from the BBR. While flux is usually the quantity of most interest for atmospheric energetics, I think for the purpose of radiative closure, performing the closure in radiance-space provides a much tighter constraint and avoids the introduction of ADM uncertainty. Some mention of the justification to do radiative closure with fluxes in the introduction of this paper would be helpful to further motivate t*he BMA-FLX product.*

The radiative closure goal of EarthCARE is defined in the MRD and it stipulates that a retrieved (cloud–aerosol–precipitation) scene with a footprint size of 10 km×10 km shall be sufficiently accurate for its atmospheric vertical profile of short-wave (solar) and long-wave (thermal) flux to be reconstructed with an accuracy of 10 Wm−2 at the top of the atmosphere. (Wehr et al, 2023, https://doi.org/10.5194/amt-16-3581-2023)

This is a very pertinent and interesting question that has been properly addressed in an independent paper published on the radiative closure in this special issue. Please visit that paper for a complete picture.

H. W. Barker, J. N. S. Cole, N. Villefranque, Z. Qu, A. Velázquez Blázquez, C. Domenech, S. L. Mason, and R. J. Hogan. "Radiative Closure Assessment of Retrieved Cloud and Aerosol Properties for the EarthCARE Mission: The ACMB-DF Product". Atmos. Meas. Tech., Special issue: EarthCARE Level 2 algorithms and data products, 2024. [Submitted]. Preprint to be available soon in the EarthCARE AMT special issue: https://amt.copernicus.org/articles/special_issue1156.html

L77-81: It is understandable that the algorithm needs to meet contingency requirements of the mission, but at the same time it does seem like a missed opportunity that the flux retrieval does not take advantage of the multi-angle views of BBR directly to constrain the radiance to flux conversion. Could this be mentioned as a potential future research activity?

Thanks for your suggestion. That approach was thoughtfully tested in the early stages of the BMA-FLX processor development. Several references included in the text ((Domenech and Wehr, 2011; Domenech et al., 2011a) describe methodologies that use the three BBR along-track radiances simultaneously for the retrieval of radiative fluxes. While the results were promising, the Agency ultimately decided to adopt a more conservative approach for the operational processor.

In any case, new approaches exploiting the multi-angular capabilities of the BBR will be explored though in later iterations of the processor. This has been mentioned and clarified in the text.

L81: I think there is an error here. The outgoing flux should NOT depend on the viewing geometry. Do the authors mean the solar geometry?

Yes, that's a typo. Thanks for spotting it. Changed to: As the outgoing flux is only dependent on the solar geometry and the radiometric properties of the atmospheric-surface domain, the three fluxes estimated by the ADMs should result in a similar flux assuming perfect instrument and retrieval responses.

L103-105: These two sentences do not make sense to me. I think I know what the authors mean, but I suggest revising the wording.

Thanks, for your comment. The paragraph has been rewritten as follows: "Scene definition refers to the classification of targets based on surface and cloud properties, as well as the angular geometry that defines the ADM model used for BBR observation. During the ADM development, the stratification of the scene definition, scene classes, determines the number of anisotropic models, which in turn dictates the number of datasets required for training the networks that construct the ADM."

L108: I do not understand why the scene type is defined separately for forward and backward scattering directions. Please clarify in the paper why this is needed.

Rewritten for clarification as: "The scene is defined separately for forward and backward scattering directions because the angular geometry significantly impacts the observed radiative properties of the target. This detailed classification is necessary because the scattering characteristics of a scene can vary significantly between forward and backward directions. By accounting for these variations, the ADM can more accurately model and predict the radiative properties of different scenes. This approach ensures that the ADM is trained on a comprehensive set of datasets, enhancing its reliability in various observational conditions."

L134-135: Using an AOD climatology might be an important source of uncertainty here. This is because the large spatio-temporal variability of aerosol means that the actual AOD for any given CERES measurement can be quite different from the climatology and therefore create a different anisotropy. Do the authors think this is important and have they considered using an aerosol retrieval product from MODIS to match instantaneously with the CERES measurements?

Thanks for your suggestion. That has been indeed considered for future developments. MODIS AOD cannot be used as part of the EarthCARE operational processing chain, however the AOD product from the MSI (M-AOT, Docter et al., 2023) is being currently being checked. To avoid introducing a new dependency the AOD climatology is still used but newer versions of the product will consider processing fluxes using NRT AOD retrievals. Discussed in the "Conclusions" section.

L143: I like the idea of using the MSI radiances directly as an input to the ANN because they contain information about cloud properties without introducing uncertainty from an intermediate cloud property retrieval. That said, I am a bit surprised by the choice of bands: 0.67, 0.865, 10.8, and 12.0 μm. The most relevant cloud properties for anisotropy are cloud fraction, optical depth, phase, and effective radius. Are the authors confident that information about these cloud properties is sufficiently represented by the 4 MSI bands that are chosen? Some supporting references would help.

Thank you for your comment. This is a very good point. The selection of MSI bands was largely influenced by the availability of bands with similar central wavelengths in the MODIS PSF-weighted radiances provided in the CERES SSF products. The authors prepared a CERES database using all daylight data from the following CERES SSF Ed4 products:

  ➢ CERES FM1, FM2 (Terra), FM3 and FM4 (Aqua) instruments in FAPS-mode between 2000-2005.
  ➢ CERES FM1 (Terra) and FM3 (Aqua) for 2007 in cross-track.

Both Terra and Aqua data was selected ensuring that the CERES instruments operated in cross-track scan mode most of the time. Cross-track scanning is optimal for developing the BBR angular models since it provides the best matching with MODIS. However, models trained using only cross-track measurements were not able to faithfully reproduce the angular geometry derived from along-track measurements. Therefore, observations from along-track scanning were also included in the database.

The MODIS bands available in the SSF Ed4 product include channels at 0.47, 0.64, 0.86, 11.03, and 12.02 microns for daylight CERES measurements. The MSI channels at 1.65 and 2.21 microns, which are important for cloud characterization, are present as additional footprint MODIS radiance statistics in the SSF Ed4 product. However, the PSF weighting is only done at clear and full CERES FOV, while the information used in the BBR ADMs includes imager statistics for both clear and cloudy areas over the BBR pixel.

The authors also analyzed data from the CERES SSF Ed3A Aqua and Terra products for the ADM development, which included the following bands:

| Spectral channels | MSI | Aqua MODIS SSF Ed3A | Terra MODIS SSF Ed3A |
| --- | --- | --- | --- |
| VIS | 0.67 | 0.65 | 0.65 |
| NIR | 0.86 | - | - |
| SWIR | 1.65 | - | 1.63 |
| SWIR | 2.21 | 2.11 | - |
| MIR | - | 3.79 | 3.79 |
| TIR | 8.80 | - | - |
| TIR | 10.80 | 11.03 | 11.03 |
| TIR | 12.00 | 12.02 | 12.02 |

All available MODIS radiances within the MSI bands were checked, and the results determined that the best combination of MSI-available radiances were the 0.65, 2.21, and 11.03 μm MODIS bands. The use of the MODIS band at 1.63 μm instead of the 2.11 μm resulted in similar ADM performance.

The 12.02 μm band was found relevant only if SWIR channels were not used. The 3.79 μm imager channel is essential for determining cloud phase. Introducing this band as input significantly improves performance, but this channel is not, unfortunately, available in the MSI design.

Paragraph rewritten as follows: The inputs for creating training sets for cloud scenes include cloud cover and radiances from the MODIS bands closest to those of MSI. The underlying assumption is that the non-linear combination of narrow-band radiances provides adequate information about the anisotropy of cloudy scenes, eliminating the need for using imager-retrieved cloud properties. Imager radiances are analyzed separately over cloud-free and cloudy parts of the observed scene. The optimal combination of narrow-band radiances includes the 0.67, 0.865, 10.8, and 12.0 μm MSI bands. This selection was primarily influenced by the availability of bands with similar central wavelengths in the MODIS PSF-weighted radiances provided in the CERES SSF products and by their importance in retrieving cloud properties using the MSI (Hunerbein et al. 2023a, Hunerbein et al. 2023b). While the short-wave infrared MSI bands significantly impact ADM performance, the CERES Ed4 product's PSF-weighted imager statistics for these MODIS bands did not include statistics for clear and cloudy areas over the CERES footprint, which are essential for constructing the BBR ADMs.

The improvement of the ADMs by using the MSI SWIR bands is foreseen and proposed for future updates. A discussion on this has been included in the "Conclusions" section.

L153-159: If I understand correctly, different datasets are used to determine the surface type for the scene identification (GLCC and X-MET) than what is used in the CERES training data (IGBP and NSIDC). Why not use the same datasets to minimize errors associated with different dataset definitions?

X-MET is the EarthCARE operational product for weather forecast data, therefore the BMA-FLX processor should use it. In the surface classification that misalignment does not really exist because the GLCC includes now the IGBP Land Cover Classification used by CERES. It is a change in the dataset name.

L232: Some more information about the ADM uncertainty is needed. Where does this uncertainty estimate come from?

Thank you for your comment. Agreed. The uncertainty estimate associated to the BBR ADMs has been now introduced in the text to better explain the merge of the flux estimates:

"The evaluation of the BBR ADMs was performed using CERES flux estimates from the database described in Section 2.1.3. BBR ADMs simulate the CERES radiative flux retrievals, which are considered the "truth" in this analysis. Notably, this dataset for evaluation is an entirely independent source, as it was not used during the training of the ADM. Following the strategy used in the training, the input parameters (NB radiances, BB radiances, cloud fraction, and surface parameters) are obtained from the CERES SSF and auxiliary products. For each ADM scene class, a flux estimate was obtained for every CERES radiance measurement, and the RMSE was computed for the entire bin. The RMSE for each scene class represents the performance of the BMA-FLX SW model compared to CERES, indicating the minimum uncertainty associated with the ADMs."

Fig. 3: The cloud optical depth would be easier to visualise with a much more limited colour bar, or maybe a non-linear colour bar. Otherwise, the vast majority of the data points are in a very limited range and it is difficult to see the variability.

Agreed, changed colour bar to logarithmic and discrete.

Figure 4: It is not very useful to plot the land cover codes and refer the reader elsewhere for the definition of the codes. Can the codes be replaced by the definitions directly in the plot? Or put in a nearby table?

Agreed, figure legend modified and added "To enhance the presentation of the results, the original land cover classes have been simplified" in the caption.

Figure 4 and 5: They are referenced in reverse order in the text. I suggest switching the order.

Corrected. Figure 4 referenced before Figure 5.

L392-393: Are the significantly worse results in Figure 6 concerning? I believe this is the most relevant uncertainty estimate because when the algorithm is applied to actual EarthCARE observations, BMA-FLX will need to use the M-CLD retrieved cloud properties, correct?

The BMA-FLX processor will indeed ingest results retrieved by the M-CLD processor. But please note that the real validation of the BMA-FLX products will take place during the Commissioning Phase, where the actual impact of MSI and ATLID products' uncertainties will be fully assessed.

L423, Table 1, and Table 2: There are several instances for both the assessment domain and the standard resolution where the combined RMSE in the SW exceeds 10 W/m2. If the RMSE is indeed the relevant quantity (see earlier question), I interpret this to mean that the radiative closure goal will regularly be

exceeded, which then leads me to question whether BMA-FLX is sufficient to achieve the closure goal of the mission. I expect that this is not the impression that the authors would like to leave the reader. I think the conclusions need some additional discussion to relate these results back to the mission requirements and what this means for the adequacy of the BMA-FLX product.

Your observation indeed points to the potential challenge in meeting the ambitious radiative closure goal of EarthCARE. However, it's important to remember that these results are part of a simulating environment, and validation with actual BBR measurements will take place in a latter stage.

The current error metrics exceeding the threshold in several instances might indicate that the radiative closure goal will not always be met. This underscore the complexity of achieving the mission's goals, and emphasizes the accuracy requirement's ambition. However, the BMA-FLX product is a significant step towards achieving them and still provides significant scientific value. The mission's goals extend beyond achieving a specific RMSE threshold to include broader scientific objectives, such as understanding radiative processes. The data obtained, even with higher uncertainties in certain scenarios, contributes to these overarching goals by providing insights into the Earth's radiation budget. The ongoing improvements to the algorithms and the comprehensive data generated in the mission will contribute to better understanding how to achieve the mission's objectives and the potential for further refinement and optimization of the BMA-FLX product.

This discussion has been added to the conclusion as follows:

In the end-to-end uncertainty assessment of the standard resolution product, the LW fluxes demonstrated strong alignment with the model truth fluxes across all three scenes, benefiting from reduced anisotropy in the oblique BBR views. The RMSEs for these scenes were consistently below 6 Wm-2. In contrast, the SW fluxes showed greater deviations from the model truth, primarily due to the more complex anisotropy of solar radiation footprints and their dependence on cloud-retrieved fields. The RMSEs for the combined fluxes varied from 7 Wm-2 in the Halifax scene to 18 Wm-2 in the Baja scene. Instances where the error metrics exceeded the 10 Wm-2 threshold suggest that achieving the radiative closure goal might be challenging, highlighting the complexity of meeting the mission's objectives and underscoring the ambitious accuracy requirements.

Despite these challenges, the BMA-FLX product represents a significant advancement toward achieving the mission's goals and provides considerable scientific value. The mission's objectives extend beyond meeting specific RMSE thresholds to encompass broader scientific aims, such as understanding radiative processes. Even with higher uncertainties in certain scenarios, the data collected contribute valuable insights into the Earth's radiation budget. Ongoing improvements to algorithms and comprehensive data generated by the mission will enhance our understanding of how to meet the mission's objectives and offer potential for further refinement and optimization of the BMA-FLX product.

It is important to note that these results are based on a simulation environment, and the proper validation of the BMA-FLX products will occur during the Commissioning Phase. This phase will involve evaluating input ingestion with actual retrievals from X-MET, BBR, MSI, and ATLID data, and will include thorough testing of flux retrieval accuracy, the parallax algorithm, and the reference level methodology.

For future improvements in the ADMs, we plan to test the use of AOD and albedo climatologies in clear-sky conditions. The inclusion of AOD climatology might introduce significant uncertainty due to the large spatio-temporal variability of aerosols, potentially causing discrepancies between the actual AOD for any given BBR measurement and the climatology, thus affecting anisotropy. We will evaluate the operational

use of the EarthCARE MSI's AOD product (M-AOT, Docter et al. 2023). Additionally, the SWIR imager channels (1.65 and 2.21 mic), crucial for determining cloud parameters, are not currently utilized in the SW ADM. Future updates will include these bands to enhance cloud field characterization. Furthermore, alternative algorithms that leverage the multi-angular capabilities of the BBR will be explored in future iterations of the BMA-FLX processor to complement the current method of integrating fluxes from each BBR telescope.

---

## Author Comment (AC3)

**Answers to Referee #2 Jacqueline Russell**

The authors thank Jacqueline Russell for her comprehensive review of the manuscript. Please find our answers (blue text) to the comments (black text) below.

**General**

The paper presents a description of the BM-FLX SW and LW radiance to flux conversions including the determination of the required cloud information from the MSI cloud properties. Using EarthCARE test frames it goes on to evaluate the error component in the fluxes due to the conversion process for both the ideal case of perfect cloud information and for the case of cloud information derived from other EarthCARE products. The work presented provides important information to users regarding the basis of the radiance to flux conversion and the evaluation undertaken is a valuable and necessary exercise providing an initial look at the flux performance compared to modelled fluxes.

The paper is generally well structured but in terms of clarity I think that some additions and modifications are needed in the description of the method in section 2. Some context to the evaluation would also help elucidate the scope and limitations of the comparisons presented. The goal of being able to estimate flux to 10 Wm-2 at the 10km by 10km for both the SW and LW is a very challenging requirement (particularly in the SW). Without knowing the details of the specification it is not immediately clear to me if this is k=1, 2 or 3 requirement on the error or an upper limit so I don't know exactly how to relate it to your results. If possible it would be good to clarify this, although I realise that these things are not always as clearly specified as they might be and may require a bit of interpretation on your part, in which case I would suggest you explain your interpretation to provide some indication of the bar you are trying to meet. I note that you mention the requirement in relation to achieving radiative closure to within 10 Wm-2 (which I appreciate is the origin in the requirement within the EarthCARE mission) but this is possibly even harder to relate to your results and to be honest I think not clearly related even at mission specification level when deriving the BBR accuracy requirement, so it may be easier to stick to discussing the flux product requirements, however this is up to you. Whatever the specifics I think it would be useful for users to be able to understand how these results indicate they should employ the data for their chosen purpose and how they might identify fluxes that meet a certain quality criteria or a means by which they may be able to meet a specific criteria. To this end it would be useful to know if quality indicators in the fluxes themselves can provide some information on accuracy of the fluxes allowing users to select an appropriate subset. Alternatively or additionally it would be useful to know if averaging over a larger domain provide generally better results (this is implicit in the bias and seems likely looking at the results and I think knowing at what scale you may meet a specific criteria would be useful). Also some discussion of how representative the errors are likely to be (given you are comparing to model fluxes), and discussion of the way forward regarding the effect of the scene identification errors would be helpful. For example, are the scene errors because these products don't meet their specifications and will be improved? Is it because the model radiances aren't as realistic as the retrieval and thus the errors are not representative of the real world or are these errors that you have to live with and can you adjust / tune the scene id to cope with them?. I don't think that addressing these things constitutes a major change to the paper in any sense. I detail these issues below along with noting some possible issues with figures 3, 5 and 6 which may be incorrectly displayed.

The work presented provides important information to users regarding the basis of the radiance to flux conversion and the evaluation undertaken is a valuable and necessary exercise providing an initial look

at the flux performance compared to modeled fluxes. The paper is generally well structured but in terms of clarity I think that some additions and modifications are needed.

**Main points – Section 2**

Lines 59 to 84. At this point in the paper you are discussing the definition of flux generally (both SW and LW), however the formulation you present is specific to the SW so is a bit confusing. I think this section should be general to both SW and LW or needs to become part of the SW section below and then be repeated for the LW. I would suggest you can generalize by stating that you are integrating the radiance field over viewing zenith and azimuth (this is true for both LW and SW). The radiance depends on the scene, the viewing geometry, and additionally for the SW the solar geometry. Note in this case you would not use relative azimuth but viewing azimuth as the integration term. You would in this case also need to avoid anything specific to the shortwave method. Alternatively, you will need to formulate a separate discussion and equation for LW and SW but I don't think this is needed.

Thank you for your comments, and agreed to keep a common formulation for both SW and LW. The text has been modified as follows: "The solar flux leaving the Earth-atmosphere ...". And it has been added "The thermal flux has the same dependencies as the solar flux except for SZA." for the LW flux for clarification. Also added at the end of the section: "In the BMA-FLX processor, the LW ADMs are assumed to be a function of only VZA, while the SW ADMs depend on SZA, VZA and RAA"

Lines 67 to 69. This is phrased a bit strangely and I am really not sure about the points these two sentences are making. Converting radiance to flux using a model of the radiation anisotropy is the method you are employing. As opposed to calculating flux from properties retrieved from radiances for example. The model used to represent non uniform variation of the radiation field (i.e. its anisotropy), is an angular distribution models (ADMs) in a general sense. You state 'The ADMs have accurately represented this variation (Su et al., 2016) and so are used as the basis for flux retrieval'. My understanding is that you are not actually directly using the particular ADMs described in Su et al. Although they may be relevant to the accuracy of the fluxes, you using this does not automatically imply any accuracy of the ADMs you derive from those fluxes if you follow a different classification system for example. If you just mean by this sentence that ADMs can be accurate, this may be true but that some angular distribution model has been used successfully before doesn't have any bearing on the accuracy of any other ADM. So I'm a bit lost and think maybe a few things are not quite said as intended, particularly bearing in mind that at this point you are talking about both the SW and LW flux derivation. I think you want to make the following points but I'm not sure if you really want to make them all here:

> - You are developing your own ADMs to represent scene anisotropy and enable fluxes to be derived from flux.
> - For the SW you are using CERES fluxes to develop your model, these employ their own distinct ADMs which are of established accuracy (although this is only relevant for the SW I think so that will need to be clarified)

Our intention here is to introduce the ADM's methodology and provide as an example of an operational ADM, the CERES ADMs from Su et al (2015).
To clarify the text, the sentence has been rewritten as: "The ADM's methodology have accurately represented this variation, e.g., Su et al. 2015, and so are used as the basis for flux retrieval. "

Line 70 to 72. I think this needs to be rephrased. An ADM doesn't estimate the flux, it represents [an estimate of] the relative angular distribution of the radiation field and can be used for the derivation of flux from a single radiance measurement. Similarly an anisotropic factor at a given set of angles does not define the ADM but can be derived from it to enable the conversion of a radiance at a given angle to flux.

Agreed, and changed to: "An ADM represents an estimate of the radiation field anisotropy and has been used to derive the anisotropic factor (R), which is the ratio between the equivalent Lambertian flux and the actual flux and enables the conversion of a single radiance measurement at a given angle to radiative flux:"

Section 2.1.3 I think it is a difficult task to try and convey your selection of inputs here via a narrative style I would suggest that a table may be an easier way to convey much of this information listing all the possible inputs the source of the information and then the scene classes where they were included. Similarly, some indication in this or a separate table of the best/chosen inputs, at present I don't think this section makes it clear what the inputs end up being. I think this would clarify a lot of what is trying to be explained and could replace much of the text in this section.

Thanks, a table has been added to clarify the inputs.

| SW ADM inputs | |
|---|---|
| MSI bands (0.67, 0.865, 10.8, 12.0 $\mu m$) | Wind speed |
| Cloud cover | Surface roughness |
| Viewing geometry (SZA, VZA, RAA) | Atmospheric gases (total-columns water-vapour and ozone) |
| SW radiances at the BBR viewing directions | AOD |
| Surface albedo | Sunglint reflectance |
| LAI for low and high vegetation | Hotspot effect |

Line 121. 'a predefined list of parameters that influence scene anisotropy' is very vague. I assume this relates to the climatology and meteorological data you speak of above. I think a table listing all the variables considered and some discussion on those eventually considered significant would solve the ambiguity.
The table has been added following your last comment.
Text changes:
"The output values represent CERES anisotropic factors, while the input vectors are a predefined list of parameters that characterize scene anisotropy (Tab. 2)."
"Additionally, climatology data on surface albedo and aerosol optical depth, along with meteorological reanalysis data on atmospheric gases, vegetation, and wind speed for cloud-free scenes, are gathered to analyze the amount of radiation reflected by the surface, the extinction of SW radiation by dust and haze, the absorption/emission of radiation by atmospheric gases, the role of vegetation in the interaction between Earth's surface and atmosphere, and the vertical wind changes, respectively.""SW radiances at the nadir, fore, and aft BBR viewing directions and the illumination/viewing geometry (SZA, VZA, RAA) from CERES are consistently included as inputs. Since SW ADMs are developed separately for each scattering direction, …"
"…we use RossThick-LiSparse geometric, volumetric, and isotropic bidirectional reflectance factor (BRDF) parameters, used to define the albedo depending on the presence of direct and diffuse components and taken from an albedo climatology derived from the MODIS MOD43B product…"

"...incident irradiance (sun-glint reflectance), given by Jackson and Alpers (2010), is also used in the training to represent radiance reflected by the ocean surface with the same angle as the satellite viewing angle."

Line 122 'BBR-received scattering direction' I'm not sure if you mean each BBR view or if you mean the forward and backwards scattering directions generally (in which case this is part of the scene) or if you mean every bin of the ADM. Can you please clarify.

It refers to the forward or backward scattering regimes as defined in 2.1.4. Sentence modified to avoid confusion to: "To determine the most significant input parameters for each class and scattering direction, we assess the variables' importance in reproducing the anisotropic factors."

Line 125 you speak of two methods which 'largely agree' how are they combined and how is any disagreement treated, do you just AND the two sets?

Sentence added: The subsets derived from the Random Forest test produced slightly better performance in ANN-based flux prediction, and, therefore, represented final subsets of ANN inputs. Clarified in the text.

Line 131 where does LAI come from? Is this a dynamic MODIS value integrated at footprint level or a generic value based on the static classification of the surface. Where do you get this information in the evaluation section it doesn't seem to be discussed.

LAI is a dynamic variable that comes from a meteorological analysis (meteorological model), as referred in the text (Poli et al., 2016).

Line 132 how do you 'consider' the hot spot effect as input, what is used as the input?

Clarified in the text as follows: "We also consider the hotspot effect as approximated in Rahman et al. (1993) as input to describe the enhanced reflectivity of the surface for an observational geometry close to the solar illumination geometry."

Line 133 to 134 you say AOD and wind speed is 'selected as parameters for network training' what do you mean by selected, is this selection the result of testing (i.e. selected by some criteria in which case please explain) or just considered to be important ('chosen')

Clarified, changed to: "Over ocean, aerosol optical depth (AOD) and wind-speed are chosen as parameters for the network training."

Line 139 how are MSI-like MODIS radiances created? Is this just the nearest wavelengths or are some spectral adjustments done, are they something you create or a defined product, can you please include further detail or a reference.

Clarified, paragraph changed to "The inputs for creating training sets for cloud scenes include cloud cover and radiances from the MODIS bands closest to those of MSI."

Line 140 it's a very plausible assumption but would benefit from an extra line of justification maybe stating which narrow bands you use and pointing out they are the ones used to retrieve cloud properties and thus implicitly contain that information albeit extracted via some other model.

It is written afterward: "The underlying assumption is that the non-linear combination of narrow-band radiances provides adequate information about the anisotropy of cloudy scenes, eliminating the need for using imager-retrieved cloud properties. Imager radiances are analyzed separately over cloud-free and cloudy parts of the observed scene. The optimal combination of narrow-band radiances includes the 0.67, 0.865, 10.8, and 12.0 μm MSI bands. This selection was primarily influenced by the availability of bands with similar central wavelengths in the MODIS PSF-weighted radiances provided in the CERES SSF products and by their importance in retrieving cloud properties using the MSI (Hunerbein et al., 2023a, Hunerbein et al. 2023b). While the short-wave infrared MSI bands significantly impact ADM performance, the CERES SSF Ed4 product's PSF-weighted imager statistics for these MODIS bands did not include statistics for clear and cloudy areas over the CERES footprint, which are essential for constructing the BBR ADMs."

Line 143 could you clarify if the process determines if these are the best or if you choose these.

Answered in the previous two questions.

Line 146 to line 147 how is the 20% of the dataset used for validation chosen and is it truly independent? E.g. every 5th ceres footprint would not be independent every 5th orbit might be, but I would suggest that 1 year out of 5 would most likely be the best test. Is the CERES validation dataset the same as the cross check validation or different? How much data is this and how is it chosen. What is the result of this validation what expected best case errors are you expected from the retrieved flux on the basis of this validation?

For model training/validation and for product validation, the team prepared a CERES database using all daylight data of the following CERES SSF Ed4 products (tens of millions of footprints):
- CERES FM1, FM2 (Terra), FM3 and FM4 (Aqua) instruments in FAPS-mode between 2000-2005.
- CERES FM1 (Terra) and FM3 (Aqua) for 2007 in cross-track.

The data was selected ensuring that CERES instruments operated in cross-track scan mode most of the time. Cross-track scanning is optimum to develop the BBR angular models since it provides the best matching with MODIS. However, models trained using only cross-track measurements were not able to faithfully reproduce measurements obtained in along-track. These models show overfitting features due to the different angular geometry of the two scanning modes. Thus, also observations from along-track scanning were included in the database

To prevent model overfitting of the ANN, i.e., when model memorizes training data rather than learning to generalize from trend, an early stopping technique is applied. In this technique the training data is randomly divided, following recommended practices for machine learning, into three subsets using uniformly distributed pseudorandom numbers. The first subset is the actual training set, where the 60% of the training data is used for computing the gradient and updating weights and biases. The second subset is the validation set (20% of the samples). The error on the validation set is monitored during the training process. The performance on the validation set start decreasing when the network begins to overfit the data. If the validation error increases for a specified number of iterations the training is stopped, and the weights and biases at the minimum of the validation error are returned. The resulting

weight and bias values together with the architecture (layer, neurons, and connections) define the network. The last 20% is assigned to the test set. If the error in the test set gives a minimum at a significantly different iteration number than the validation set error, this indicates a poor data selection for the training set.
The BMA-FLX fluxes, the product, are also validated against another independent dataset randomly chosen from CERES database. The results of this validation are used to set the uncertainties for each of the classes of the ADM scene definition.

Line 155 You say the retrieval algorithm uses two surface types and combines them, do you mean you make two separate retrievals for each class individually which are then combined or do you mean there is some sort of combined class possible in the retrieval?

There are two different scene classes, one ADM per each. Thus, there's one anisotropic factor per each. Then the anisotropic factors are used, along with the corresponding surface coverage and albedos, to compute a mixed anisotropic factor. Then a single flux with the mixed anisotropic factor is retrieved. Agreed, and changed to: In clear-sky scenarios, the retrieval algorithm employs the two surface types with highest coverage within the BBR pixel, defining the observation as a mixed scene. Two ADM scene classes are selected, obtaining two SW anisotropic factors for the footprint. The pure anisotropic factors obtained for primary and secondary surface areas within the BBR fore, aft, and nadir observations are combined into a mixed anisotropic factor (Bertrand et al., 2005), weighting the pure anisotropic factors by their respective (scaled) surface coverages and TOA albedos. The mixed flux is then calculated using the corresponding SW radiance and the mixed anisotropic factor."

Line 162 what DEM is used, and do you also do this for elevations below 0 (assuming zero is the average Geoide or is it something else)? Although I'm a bit confused because surely the coregistration at the surface performed by BBR considers some sort of DEM already.

The DEM used is a post-processed version of the ACE-2 DEM common to all the processors in the EarthCARE mission. Berry, P. A. M., R. Smith, and J. Benveniste. 2010. ACE2: The New Global Digital Elevation Model. In: Mertikas, S. (eds) Gravity, Geoid and Earth Observation, International Association of Geodesy Symposia 135: 231-237. Berlin, Heidelberg: Springer. https://doi.org/10.1007/978-3-642-10634-7_30.
In the BBR L1 the three views are co-registered at altitude 0 in a geodetic reference frame, but the L2 BMA-FLX processor takes into account the DEM for the co-registration of the views. For example, the co-registration is done at the surface elevation for clear-sky conditions.
Added reference to the paper: "In clear-sky conditions, without cloud parallax and at sea-level locations, the default surface co-registration is used. However, if a new co-registration is needed (i.e., surface elevation is greater than 0 and/or a cloud is observed in the nadir view, see section 2.1.5), the ADM scene classes for the BBR oblique observations are reconstructed. When the digital elevation model (DEM, Berry et al. (2010)) indicates elevations above 0, the scene identified for the nadir view is also used in the BBR oblique views in the new co-registration."

Line 164 to 166. This sentence is unclear at this point, it becomes clearer after figure 1 is discussed maybe reference this later discussion or consider rearranging such that the 2.1.5 comes before 2.14? Also I would replace cloud properties with cloud fraction as I think this is the only 'property' you are using for scene? If you are also using other properties for input it might be worth clarifying somewhere what is used for that for the oblique view although probably this shouldn't go in the scene section, although I'm

not sure it would make sense to use nadir properties in that case unless that somehow corresponds to what was used in the training.

Reference to section 2.1.5 added and rewritten as "To reconstruct the scene classes for the oblique views in cloudy nadir conditions, the cloud fraction observed in the nadir view and the surface definition from the displaced oblique views are used (new fore and aft BBR views intersecting the cloud top observed in the nadir view, see section 2.1.5)."

Line 166 to 175. I am confused by the discussion on cloud top height in this section on scene identification as whilst it has been described as an input it has not been discussed as part of the scene classification. Is this part of the scene classification (in which case it should have been discussed in scene definition section), does it just relate to matching scenes (in which case can you clarify this) or is this really a discussion on inputs in which case it should go elsewhere or the scope of this section should be extended.

Indeed, it is not part of the scene classification. However, it is needed to reconstruct the scene in the presence of clouds in the nadir view. To clarify it the text has been rewritten as: "To reconstruct the scene classes for the oblique views in cloudy nadir conditions, the cloud fraction observed in the nadir view and the surface definition from the displaced oblique views are used (new fore and aft BBR views intersecting the cloud top observed in the nadir view, see section 2.1.5). The cloud mask and cloud top height, used to match the views, are derived from the MSI operational retrievals (Hunerbein et al. 2023b)".

Line 183 'provided there is no significant elevation' do the BBR products coregister at the 'surface' or at some average sea level, surely the former which would include elevation of surface?

Answered in the response to comment above "Line 162".

Line 195 to 196 'CTH…is a reliable estimator for co-registering…in the thermal' was this shown in the AATSR study or some other way how is this know. One might have thought that thin cloud may pose a similar issue in the thermal as in the visible, or that wet atmospheres might may the co-registration point above a low cloud, or in clear sky make the surface a poor point of coregistration.

The AATSR study (see BMA-FLX ATBD 7.3. Coregistration algorithm) has shown that, in general, the CTH is a reliable estimator for co-registering the LW, but it could probably be improved for specific scenes. The verification of the Reference Level is foreseen during the commissioning phase.
Please note that the EarthCARE L2 ATBDs are not yet published, but they are expected to be released during the next months.

Line 202 how are the errors in the ADM treated when minimizing the height of co-registration, what happens in cases that are poorly behaved and a minimum isn't found?

The flux errors are not considered (nor modified) during the minimization technique. First the fluxes are calculated with the ANN and have their uncertainty (ADMs + BM-RAD). Then the reference level is found with the minimization technique. The flux (with its associated error) in that reference level is considered, but there is no more error calculation involved.

Lines 223 to 225. I find this sentence difficult to understand. I think maybe don't use the word 'coordinates' unless you really mean 2d or 3d coordinates and or specify what coordinates you are talking about. So if you just taking about the central pixel in the nadir view, say this, if you are talking about the 3d location of the nadir reference level then specify this, if you mean the location of the ctp in the nadir then again specify.

Agreed, changed to "Given a tropopause height, $h_T$, the oblique observation is considered affected by cloud parallax if the CTH of clouds along the optical path is located at a height between the surface elevation of the nadir clear-sky domain (or CTH of clouds of a cloudy nadir domain) and the tropopause, the cloud height being $h_i = d_i/\tan(\theta_{obl})$, where $d_i$ ranges from 1 to $d_{maxparallax} = h_T \tan(\theta_{obl})$."

Line 228 before equation 5. Do you calculate this for each combination, i.e. nadir vs fore, nadir vs aft and aft vs fore?

Done for each combination (nadir-fore, nadir-aft, fore-aft) and then compared to the fractional error threshold (10%) to see how many flux estimates agree/disagree. If the three flux estimates disagree, the one with lower ADM error is considered. If at least one agrees, the ones who disagree are not considered for the final flux estimate, which is the combined flux. This is written below, lines 234-235.
Text updated for clarification: "In absence of parallax, the discrepancies between the fluxes derived from each combination of nadir, aft, and fore observations are calculated as follows"

Line 229 this equation does not necessarily limit the range of 0 to 200%, if you limit it artificially maybe this should be specified in the equation by an alternative formulation, i.e. as stated if < 200% and 200% if greater than that.

Please note that the Fractional error $= 100/(1/2)*|Fy-Fz|/(Fy+Fz)$.
The equation limits the range to 0-200% by definition. For instance, under these three different situations:
- In an ideal case where all fluxes are the same, then the fractional error is 0%.
- If are not equal but one is zero, then the fraction $|Fy-Fz|/(Fy+Fz)$ is 1 and the fractional error is 200%.
- If are not equal and none is zero, as fluxes are always positive (and in case of invalid are fill values, which are positive numbers), the fraction $|Fy-Fz|/(Fy+Fz)$ is always <1. Thus, the fractional error is <200%.

Line 244 do you have any evidence that using a set of regressions solve the problem and if so why/how. Maybe you need to just state the cause of the issue was a deviation from the fit used for this scene type (rather than an issue for a plane parallel assumption for example which would not be solved but a different regression). Also is this set just the splitting into 5 degress in VZA and 20 Wm-2sr-1 bins or is it something more based on cloud information could you explain somewhere in this paragraph, either state on line 274 that it is radiance and angle separated or explain it is also cloud property dependent and include this addition on line 275.

Thanks for your comment, indeed the set of regressions itself doesn't solve the problem in the case of semi-transparent clouds over warm surfaces. However, regressions using brightness temperature differences in the split window channels does. During the development of the LW algorithm, 9 regression models were evaluated, being the one used in the BMA-FLX processor selected as the model that provided better results for the estimation of the thermal fluxes. The text has been rewritten for

clarification as: Different regressions, and consequently different anisotropy models, have been developed for thermal radiances in bins of 20 W m2 sr-1 and also every 5 degrees in VZA.

Line 265 how are uncertainty estimates associated with the fluxes derived this was derived, are they related to the MSI BT uncertainties propagate through the regression used to derive the anisotropy? Do they take into account the scatter about the original regression?

The MSI BT uncertainties were not available to propagate the errors for this study in which simulated data is used. So far the theoretical error in the calculation of the anisotropic factor is taken into account. Evaluation of the error in the flux calculation has been done via evaluation derived fluxes in the SEVIRI disk and then compared against CERES data using collocated CERES and BBR-like/SEVIRI data. In this study the results showed that for single view flux retrieval the error at one standard deviation is about 6 $Wm^{-2}$ , and for the combination of the three views it is reduced to 3 $Wm^{-2}$.

Line 267 can you state what the reference level is in clear sky, is it the surface or somewhere above and does it vary with atmospheric moisture?

In section 2.1.5 (lines 181-183) it is mentioned that in clear-sky conditions the co-registration is performed at surface level: "BBR radiance measurements are typically co-registered at surface level by default. In clear-sky conditions, the primary emission or reflection observed is from the surface, making this default radiance collocation adequate, provided there is no significant elevation. "
It does not change with atmospheric moisture.

Line 280 how it is evaluated from the CERES data? I assume you mean that the ceres data indicate that equal weight should be given to the forward and nadir views in combination. It is not clear what you are doing here but I suggestion (see the following point) that it might be worth a bit of explanation.

True, some context is missing here. The estimation of the best alpha value was done in a previous study using all the available CERES FM2 True-Along-Track (TAT) data (6 days from February 2005). In this study, a flux from the Direct Integration (DI) of LW radiances along the track has been calculated and compared to a linear combination of the CERES fluxes for each BBR view (+/-55, nadir). This database of true-along-track data included 913,064 km of valid orbit data. The best linear combination of the 3 views resulted in weighting factors very similar for the fore and aft views and slightly lower for the nadir view: $F\_DI = 0.3467 \ F\_fore + 0.3424 \ F\_aft + 0.3089 \ F\_nadir$.
The fore and aft views were not biased and RMS differences with respect to the DI flux were about 6.4 W $m^{-2}$. The nadir view was slightly biased but RMS difference with the DI flux was a bit smaller 5.8 W $m^{-2}$. Details of this study can be found on the ATBD section 7.2, normally to be released to the public after commissioning.

Line 281 I think this requires a bit more comment. You are saying the studies with CERES implies that the LW radiance distribution doesn't behave in a plane parallel way. However you are using plane parallel assumptions to base your radiance to flux conversions on so this would seem potentially concerning. If your comparison with CERES is based on applying your theoretical plane parallel ADMS to forward and nadir and comparing the individual vs combined result with the CERES empirical ADM conversion then this would support this equal weighting to more closely match the empirical CERES result. But I think this needs explaining and possibly the actual improvement and discrepancies found stated here.

Related to previous question, it has been rewritten as: "The best value for the α parameter was evaluated in a previous study using all the available CERES FM2 true along-track data (6 days of February 2005). In this study, a flux from the Direct Integration (DI) of LW radiances along the track has been calculated and compared to a linear combination of the CERES fluxes for each BBR view (fore, nadir, aft). The best linear combination of the 3 views resulted in weighting factors very similar for the fore and aft views and slightly lower for the nadir view being α very close to 1/3."

**Main points - Section 3**

Lines 325 to 329. I think it would be helpful to have a previous section clarifying the two step comparison you will execute in 3.2 and 3.3 and the purpose and scope of the two comparisons and strengths/weakness/limitations of these assessments. I think the two step approach you use is reasonable (and in fact wonder why you don't go further and substitute a 'model truth' for the X-MET data. But I think it would be useful to also discuss the limitations and issues of the model truth in the context of your method as whilst it assesses much of the performance it will obviously be limited to how the model fluxes relate to the ADMs which is not necessarily going to be the same as the real world. You discuss some of this such as surface missmatch when you consider differences but I think it might be helpful to have a broader stage setting consideration of how the model fluxes might not match the SW adms you use, or how assumptions in the longwave such as cloud modelling or surface emissivity might differ between the model the information used to derive your ADMs.

Thank you for your comment. We have added the following introduction in section 3 to better clarify the purpose, scope, strengths and limitations of this evaluation:
To assess the robustness of the BMA-FLX processor, we implemented a two-step evaluation process. The primary objective is to evaluate the expected accuracy and reliability of the BMA-FLX processor in retrieving radiative fluxes using data from simulated EarthCARE orbits. Proper validation of the fluxes retrieved by the BMA-FLX processor is anticipated during EarthCARE's Commissioning Phase. However, these two steps are designed to isolate and analyze different sources of uncertainty and error in the flux retrieval process. Section 3.2 describes the processor uncertainty assessment using model truth cloud profiles and model truth snow and sea ice properties. This step evaluates the intrinsic performance of the BMA-FLX processor by eliminating uncertainties introduced by Level 2 cloud retrieval algorithms from other instruments. Here, input data derived from MSI and ATLID's forward models are replaced with data directly taken from geospatial simulations. By using model truth cloud and snow information, this assessment focuses on the processor's ability to handle ideal input conditions without compounded errors from preceding algorithms. Section 3.3 presents an assessment of end-to-end uncertainty using operational L2 products. This step evaluates the performance of the BMA-FLX processor under realistic conditions, where input data include the uncertainties and errors from the operational L2 EarthCARE processors. The BMA-FLX processor ingests L2 products derived from simulated L1 data, simulating an operational scenario. This provides insights into the processor's anticipated issues and robustness in the presence of non-ideal inputs.
The isolation of processor performance using model truth data allows for the identification and resolution of intrinsic issues without external retrieval errors. In contrast, the realistic operational conditions incorporate all sources of uncertainties from the operational L2 products, providing a comprehensive evaluation of the processor's performance in simulated mission conditions. The main limitations of this evaluation include reduced scene diversity and potential discrepancies between the radiative transfer simulations used in the simulated geophysical data and the modeled EarthCARE products. The test frames are based on three specific EarthCARE orbits, which, although diverse, may

not encompass all possible atmospheric and surface conditions encountered globally. Additionally, discrepancies in surface definitions between the RTC model and the BMA-FLX processor can affect flux retrieval accuracy, particularly in clear sky conditions.

Line 335 'being 0.1 the one with better performance' → '0.1 being found to have the best performance'. I don't understand what is being done here, what determines the best performance? Also why is this not needed to be repeated for the retrieved parameters, this would account for any offset in the retrieval. Will this be redone operationally to determine an operational threshold and if so how will performance be assessed in that case?

The best performance was meant to indicate the case in which the best metrics in the comparison of the resulting fluxes with the model truth fluxes is found. For clarification the text has been updated as follows: Different COT thresholds were considered to calculate both the cloud mask and the cloud top height, 0.1 being found to have the best metrics in the analysis of the flux results".
The M-CLD processor is responsible for the cloud properties retrieval and it is being developed by a different team. During the commissioning phase, it is expected that the processor dependencies are further explored, but we cannot confirm that this will be redone operationally.

Figure 3 – Upper panels please label the colour bar and check that it is correct. The caption states it is cloud optical depth (at 680nm ?) but the scales seem quite extraordinary. I would suggest if you really have COT up to 800 you might consider a logarithm scales as a more useful way to provide information in these figures.

Yes, it is COT at 680 nm (it is indicated in the "Processor uncertainty assessment" section). The upper panels colour bars have been modified (discrete and logarithmic) as suggested.

Lines 342 to 353 I think that results in table 1 need to be discussed in terms of the differences between the simulations and the FLX processor assumptions. For example the SW flux retrieval is based on observed radiance distributions, albeit somewhat indirectly (via CERES fluxes based on observationally derived ADMs), in contrast to plane parallel (?) model calculations, can this be an explanation of the variation in the sign of the bias between views in the SW do you think? What might explain the consistent offset in the LW which actually seems worst in the nadir view, is this a consistent difference between the limb darkening effect between the model simulation and your simulations or a surface emissivity effect or something you might expect to persist with real data?

That is an interesting point to analyze, however we cannot conclude with the available results that the change in the sign of the bias between views in the SW is related to the fact of using RT plane-parallel test scenes. In the LW we would expect the same behavior with real data, in which the estimation of the flux only from the nadir view has slightly higher error than the off-nadir views. For the LW has been rewritten as follows: As expected, given the lower anisotropy of the radiance field for the off-nadir views (Suttles et al., 1989), the aft and fore views present the lowest differences with respect to the model truth in the LW fluxes

Line 358 Are you saying you use only the GLCC classes and don't use anything equivalent to the X-MET to id fresh snow, later (line 402) you seem to imply you use X-MET would it not be helpful to put what you use in the ADM here if this is different, maybe you do and you just need to clarify you use GLCC updated where appropriate with X-MET here?

The IGBP classes that come from BM-RAD are related to the Global Land Cover Characteristics database (GLCC). These IGBP types are simplified to 7 permanent classes. Two dynamic classes, sea-ice and fresh snow, are added to the surface classification and are derived from the snow depth and sea-ice cover parameters of the X-MET data. As discussed in section 2.1.4.

Changed for clarity to: The cloud cover from the model truth (ranging from clear to overcast) and the surface classification from GLCC and X-MET, both parameters employed in the ADM scene identification (e.g., water bodies, forests, savannahs, etc.), are presented at the bottom of the plots. GLCC database provides a surface classification of seven permanent classes. Two dynamic classes, fresh snow and sea-ice, are derived from the snow depth and sea-ice cover parameters of the X-MET data and added to the classification.

Line 367. I think before you are too hard on your performance under broken cloud conditions you should consider how well these are likely to be simulated. The 'complex interaction' seems to imply a 3d effect but are these considered by the model truth? Or do you mean to say this might be a reference level issue?

Indeed, in plane-parallel RT models, broken clouds are likely to be poorly simulated due to the model's assumptions of horizontal uniformity. These models might fail to accurately depict the varying optical properties of the scattered clouds. As a result, the SW fluxes, which are obtained from ADMs constructed using satellite measurements, retrieved in the regions with simulated broken clouds tend to be noisy and less reliable.
Text modified accordingly: In plane-parallel RT models, broken clouds are likely to be poorly simulated due to the model's assumptions of horizontal uniformity. Consequently, SW fluxes, which are obtained from ADMs constructed using satellite measurements, retrieved in the regions with simulated broken clouds tend to be noisy and less reliable. In the LW estimates, this results in noisy flux retrievals that are flattened out when increasing the averaging region. Thermal fluxes obtained for the assessment domain resolution (5 JSG x 21 JSG pixels) smooth the response, which contributes to the success of the radiative closure.

Line 370 to 374 It is not immediately obvious which has snow and which doesn't in your assumption vs the model truth, but I think you are saying you don't assume snow because it isn't in the X-MET data but the model truth includes snow. Have you confirmed that if you use snow here that the problem goes away. Is using the X-MET data sensible in this case rather than using the model truth surface?

The main reason for the discrepancies was the difference between the fresh snow and sea ice information provided by X-MET and the snow/ice information used in the RTC simulations. We tested this by using X-MET mock-ups that incorporated the fresh snow and sea ice data utilized in the RT calculations. However, the results were not ready for publication at the time of manuscript submission. We have now included this new analysis in the manuscript, and as expected, the results show significant improvement.

Figure 5 Although the mean lines look they are probably correct, the Std and RMSE plotted on the graphs don't seem to be correctly plotted or at least do not correspond to the values shown plotted around the mean. Furthermore, I think it would be useful to see how the errors relate to any quality indicators on the flux you have (e.g the flux uncertainties from the ADM (equation 6) you have for the SW and or the

discrepancy between the fluxes derived from different views or some other measure from the retrieval if it is good or not). Maybe this could be indicated by colour coding the different points that fell above a certain retrieval goodness threshold or by including an additional plot of flux difference against retrieval uncertainty.

That's correct, thanks. We had a bug in the plotting function, corrected and updated in Figs. 5 and 6. We agree that the idea of showing flux uncertainties in the plots is indeed interesting, but it would not be easily appreciated and could led to the misinterpretation of the results.

Figure 5 Also in the longwave there seems to me to be an indication of noise in the retrieved flux that might be significantly improved if you increased the averaging region (smooth the results seen here). Do you think this is the case. Would it be worth looking at what improvement this brought to the percentage of points within the 10Wm-2 line, and would be useful information in the context of closure style comparison studies.

These noisy values smooth out when increasing the averaging region, as seen in the plots below where the pixel size corresponds to the Assessment domain (5x21 km2). This is the BBR spatial resolution used to perform the radiative closure, as explained in H. W. Barker, J. N. S. Cole, N. Villefranque, Z. Qu, A. Velázquez Blázquez, C. Domenech, S. L. Mason, and R. J. Hogan. "Radiative Closure Assessment of Retrieved Cloud and Aerosol Properties for the EarthCARE Mission: The ACMB-DF Product". Atmos. Meas. Tech., Special issue: EarthCARE Level 2 algorithms and data products, 2024. [Submitted]. Information added to the paper.

[Figure]

[Figure]

Lines 384 to 387. You discuss the differences between the cases but is there indication of why they behave so differently?

Thanks for your comment. This is already explained in the last paragraph of the section. However, for clarity, we have introduced brief explanations of the behavior in the paragraph where the results are discussed. Please see below the update:

In SW, the impact of using cloud properties retrieved by M-COP and M-CM is minimal in the Halifax scene. However, in the Hawaii scene, the flux results for the aft and nadir views are significantly higher compared to the analysis using model truth cloud fields (Table 3). This discrepancy arises from the MSI's cloud mask retrieval failing to accurately detect the cloud fraction in the tropical convective system at the center of the Hawaii simulated orbit. Selecting an incorrect SW ADM for very bright cloud scenes significantly impact in the average metrics. The combined results for Baja do not differ significantly from the previous analysis. Nevertheless, flux results for fore and nadir views exhibit greater uncertainties compared to those obtained with the true cloud profiles. This is primarily due to the challenges faced by the M-CLD processor in retrieving cloud information from the simulated cloud fields over snow surfaces above 50° N.

[Figure]

[Figure]

[Figure]

[Figure]

[Figure]

[Figure]

Line 387 Is the lack of cloud information from M-CLD just for this test or is it going to be true operationally? What is the implication if the latter?

The discussion of the M-CLD processor's performance for the three EarthCARE test scenes is presented in:

Hünerbein, A., Bley, S., Deneke, H., Meirink, J. F., van Zadelhoff, G.-J., and Walther, A.: Cloud optical and physical properties retrieval from EarthCARE multi-spectral imager: the M-COP products, EGUsphere, 2023, 1–23, https://doi.org/10.5194/egusphere-2023-305, 2023.

Hünerbein, A., Bley, S., Horn, S., Deneke, H., and Walther, A.: Cloud mask algorithm from the EarthCARE Multi-Spectral Imager: the M-CM products, Atmospheric Measurement Techniques, 16, 2821–2836, https://doi.org/10.5194/amt-16-2821-2023, 2023.

The analysis of the processors' inputs accuracy for the entire L2 processing chain is expected to take place during the EarthCARE Commissioning Phase.

Lines 394 to 401 What are the implications of this, is it expected that M-CLD will improve or can/should some adjustment be made to the result before employing them in this manner? Or are these results a truer representation of the FLX accuracy and indicate its reliance on M-CLD c.f. that used in the classification? Is the 0.1 optical depth threshold found to perform best earlier still valid or does it need to be adjusted?

It is important to note that the paper's results are based on a simulation environment. Proper validation of the BMA-FLX products will occur during the Commissioning Phase, where the ingestion of actual retrievals from BBR, MSI, and ATLID data will be evaluated. Once M-CM product becomes available, the BMA-FLX performance will be assessed using the default M-CLD retrievals. These results will be discussed

with the M-CLD team to determine if any adjustments are necessary, including the potential reassessment of the cloud optical depth threshold.

Line 393 You state the results are significantly worse, they seem to have a lot of scatter to them and I wonder to what extent they might improve if you increased the averaging domain (possibly because the cloud information is more accurate on these scales) or is this a result a biases in the cloud information. Can you relate this to known errors in the cloud information, do they meet their requirements are they expected to improve in orbit or be improved.

Question discussed in Figure 5 comment.

Figure 6. As was the case for figure 5 although the mean lines look to correspond to the stated values the lines indicating the RMSE and Std don't seem to be correct. The same points about showing the errors in the context of retrieval uncertainty main in relation to figure 5 apply here, it would be good know if you can tell you are likely to have poorly retrieved fluxes.

See answer for question related to Figure 5.

**Main points – Section 4**

Can you clarify how your results relate to the goal requirements, I realize this may require some interpretation on your part. I don't think that the std can be related to the 10Wm-2 requirement unless you also consider the effect of the bias. Can points meeting the requirement be identified via a flux quality indicator or by excluding some subset of scenes or by averaging over a larger domain if so what size. Can you add some discussion of what happens next, if the scene id is expected to improve in orbit, if it needs to be altered or if you can tune your flux retrieval.

- Thanks for your comment, the discussion of the conclusions has been modified as follows: In the end-to-end uncertainty assessment of the standard resolution product, the LW fluxes demonstrated strong alignment with the model truth fluxes across all three scenes, benefiting from reduced anisotropy in the oblique BBR views. The RMSEs for these scenes were consistently below 6 Wm−2. In contrast, the SW fluxes showed greater deviations from the model truth, primarily due to the more complex anisotropy of solar radiation footprints and their dependence on cloud-retrieved fields. The RMSEs for the combined fluxes varied from 7 Wm−2 in the Halifax scene to 18 Wm−2 in the Baja scene. Instances where the error metrics exceeded the 10 Wm−2 threshold suggest that achieving the radiative closure goal might be challenging, highlighting the complexity of meeting the mission's objectives and underscoring the ambitious accuracy requirements.

    Despite these challenges, the BMA-FLX product represents a significant advancement toward achieving the mission's goals and provides considerable scientific value. The mission's objectives extend beyond meeting specific RMSE thresholds to encompass broader scientific aims, such as understanding radiative processes. Even with higher uncertainties in certain scenarios, the data collected contribute valuable insights into the Earth's radiation budget. Ongoing improvements to algorithms and comprehensive data generated by the mission will enhance our understanding of how to meet the mission's objectives and offer potential for further refinement and optimization of the BMA-FLX product.

- There is a flux quality status and the estimated error in the flux retrieval available in the BMA-FLX product for both solar and thermal TOA fluxes/per view and also for the combined flux. The idea of using different assessment domains will be explored during the commissioning. As a starting point, it has been chosen as 5x21 JSG pixels.
- Added to the discussion:

  For future improvements in the ADMs, we plan to test the use of AOD and albedo climatologies in clear-sky conditions. The inclusion of AOD climatology might introduce significant uncertainty due to the large spatio-temporal variability of aerosols, potentially causing discrepancies between the actual AOD for any given BBR measurement and the climatology, thus affecting anisotropy. We will evaluate the operational use of the EarthCARE MSI's AOD product (M-AOT, Docter et al. (2023)). Additionally, the SWIR imager channels (1.65 and 2.21 μm), crucial for determining cloud parameters, are not currently utilized in the SW ADM. Future updates will include these bands to enhance cloud field characterization. Furthermore, alternative algorithms that leverage the multi-angular capabilities of the BBR will be explored in future iterations of the BMA-FLX processor to complement the current method of integrating fluxes from each BBR telescope.

**Minor corrections**

Line 3 design → designed

Thanks, fixed: The satellite's payload include four instruments designed to synergistically retrieve vertical profiles of clouds and aerosols, along with the atmospheric radiation data.

Line 4 remove 'an algorithm' (it is a processor specifically created I assume consisting of several algorithms)

Thanks, 'an algorithm' has been removed.

Line 6 to 7 'measurements' It is not clear here if you are talking about the radiances or the fluxes or all the products. It might be clearer to change measurements to radiances if that is the intent, radiance and fluxes of simply all products depending on what you mean. / Lines 7 and 8 'of the atmosphere in cloud condition (reference level)' doesn't make sense. Do you mean '..or in cloudy conditions at a reference level which corresponds to the radiatively most significant vertical layer of the atmosphere'

Thanks, rewritten as: These radiances are co-registered either at the surface or, in cloudy conditions, at the radiatively most significant vertical layer of the atmosphere (reference level)

Lines 12 to 14. 'The radiance to flux conversion algorithms have been successfully validated.....' Make clear this is done here in this paper and state the result obtained in the paper, something along the lines of. 'Validation of the radiance to flux conversion through end-to-end verification using L1 and L2 synthetic data for three EarthCARE orbits. The results .....'

Thanks for the suggestion. Sentence added: In general, a good agreement is found between the retrieved fluxes and the model truth, with RMSEs varying between 7 W m$^{-2}$ and 18 Wm$^{-2}$ for the solar fluxes and lower than 6 Wm$^{-2}$ for the thermal fluxes.

Line 40 (and multiple other occurrences) change 10x10km2 → 10km x 10 km throughout. Could also be 10 by 10 km or square with side of 10km or 100km2 region (it is 100km2 but you are not presenting the equation so I think 10x10km2 is a bit strange. Wehr et al 2023 used 10km x 10km so this might be safest.

Thanks, corrected.

Line 44 'are challenging' → is challenging (challenging relates to estimation which is singular)

Thanks, changed to: Due to the highly anisotropic character of some physical phenomena, like the reflection of solar radiation by clouds, the estimation of the radiative fluxes from measured radiances at a single Sun-observer geometry is challenging.

Line 50 'is created per scene and constructed from'. I'm not sure if you are trying to say that the algorithm is determined from scene stratified observations from CERES and MODIS, or it you are saying that it is a scene dependent algorithm or both, could you please rephrase to clarify.

Fixed, changed to: The BMA-FLX SW algorithm is scene dependent and has been constructed from six years of Clouds and the Earth's Radiant Energy System (CERES) and Moderate Resolution Imaging Spectroradiometer (MODIS) Terra and Aqua measurements using an artificial neural network approach.

Line 57 'the mission goal of 10 Wm-2' please explain what the goal is.

Sentence added to give more information about the goal: The accuracy requirement placed under the EarthCARE radiative closure's goal is defined in the EarthCARE mission Requirements Document (MRD) (Wehr, 2006).

Line 62 'integrating the radiance field' need to add somewhere in the sentence that the integration is over viewing and azimuth angles.

Thanks, rewritten as: "The solar flux leaving the Earth-atmosphere system is obtained by integrating the radiance field, $L(\theta_0, \theta, \phi)$, over the solar zenith angle (SZA, $\theta_0$), the viewing zenith angle (VZA, $\theta$), and the relative azimuth angle between the Sun and the satellite view (RAA, $\phi$) as follows:"

Line 76 could you state the time period during which the 3 views are obtained to clarify the 'almost simultaneously'. Also I would suggest replacing 'providing a detailed view of scene anisotropy' to 'providing information on scene anisotropy' as whilst three views are helpful, they are not necessarily 'detailed' information.

Rewritten as: The BBR instrument observes each target on Earth from three different directions almost simultaneously (about 3 minutes between the fore and aft views), providing information on scene anisotropy.

Line 95, 'CERES instrument' → 'CERES data' (assuming you are using a data product) and state the version of the CERES product used and the years used.

Thanks, changed to: These studies are further developed in the BMA-FLX SW processor, where the radiance-to-flux conversion algorithm employs a feed-forward backpropagation ANN to model the ADM-

based fluxes from the CERES Single Scanner Footprint TOA/Surface Fluxes and Clouds (SSF) Edition 4 product (Su et al., 2015)

Line 104 'scene definition concludes the number of ANN training sets' I don't think that concludes is the correct word here, do you mean determines?

Thanks for your suggestion, changed to: Scene definition refers to the classification of targets based on surface and cloud properties, as well as the angular geometry that defines the ADM model used for BBR observation. During the ADM development, the stratification of the scene definition, scene classes, determines the number of anisotropic models, which in turn dictates the number of datasets required for training the networks that construct the ADM.

Line 106 'relies on' → 'consists of' / Line 106 to 107 You say there are 6 static surface types but list 7 in the brackets, so either the number stated or the listing is incorrect / Line 106 'type' → 'types'

Fixed. / The number was wrong. Changed "**six**" to "**seven**". / Fixed.
Changed to: Our scene definition consists of seven static surface types (ocean, forest, savanna, grassland, shrub, desert/ bare soil, and permanent snow), plus two dynamic ones (fresh snow and sea ice), and four cloud fractions, CF (cloud-free, partly covered 0.1 < CF < 50, mostly covered 50 <= CF < 99, and overcast), taking into account that for overcast conditions the categories forest, savanna, grassland, shrub, and desert are grouped in a new category named land.

Line 108 can you somewhere in this sentence confirm the total number of scene types (I assume this is 72 but it would be good to state.

Agreed, added Table 1 with Scene definition for clarification, and the following sentence: This classification is done for each scattering regime (nadir, forward, and backward), resulting in a total number of 96 scene classes presented in Table 1.

| Scene definition | |
|---|---|
| Surface type | Sky condition |
| Land | Overcast |
| Forest
Savanna
Grassland
Shrub
Desert/ Bare soil | Clear-sky, partly covered, mostly covered |
| Ocean
Fresh snow
Permanent snow
Sea ice | Clear-sky, partly covered, mostly covered, overcast |

Line 110 Are the NSIDC and NESDIS snow maps combined by you or are they combined by the SSF product? If the latter then change the sentence to say you use the CERES SSF snow information which is a combination of these, if the former then explain how you combine them.

We used the snow information coming from the CERES SSF product. Changed as suggested: The static categories considered for training are obtained from the International Geosphere–Biosphere Programme (IGBP) land cover, while the fresh snow and sea ice surface types are derived from the CERES SSF snow information as a combination of the microwave snow/ice map from the National Snow and Ice Data Center (NSIDC) and the snow/ice map from the National Environmental Satellite, Data and Information Service (NESDIS).

Line 137 'considered' → 'included'

Fixed, changed to: To further consider anisotropy changes due to atmospheric characteristics, we also included the total-column water-vapour and total-column ozone from reanalysis.

Line 148 'estimates and original'…'estimates compared to the original…'. Do you have this estimate?

Yes, those estimates are computed to obtain the uncertainties of the ADMs. Fixed as: The root mean square errors (RMSE) of the ANN-based flux estimates compared to the original CERES derived fluxes define the theoretical uncertainties for each scene class.

Line 152  you mention fore and aft BBR observations, but presumably it also classifies the nadir?

No, only off-nadir views are classified as either forward or backward.

Line 189 surely it is 'fore and nadir' from AATSR not 'nadir and aft', the along track view is in the forward direction and obtained before the nadir view.

Thanks for spotting the typo. Corrected: Flux measurements for nadir and fore observations across several orbits of the Advanced Along-Track Scanning Radiometer (AATSR) were calculated using a radiance narrow-to-broadband conversion.

Line 190:  can you include a reference or further detail of the NB to BB used at least the channels employed.

AATSR bands specified, and corresponding references added (Clerbaux et al. 2005) in the text:  The SW NB to BB used the AATSR solar channels 0.6, 0.8 and 1.6 μm (Clerbaux et al., 2005),  while the LW used the thermal channels 10.8 and 12.0 μm in a second order regression.

Line 208 equation 4, is there a reason that there is no vza dependence indicated in R and L?

There's the index "i" representing the view, which is the vza dependent variable.

Line 214 'ideally reflected from the same atmospheric domain' what does this mean 'ideally' and 'domain' specifically.

Rephrased for clarity as: The oblique views that minimize the flux differences are displaced by a distance d from the default surface co-registration. These oblique observations, which are expected to be

reflected from the same atmospheric region, are then co-registered with the nadir observation of sample i.

Line 219 what 'internal consistency check' can you please explain what is done, is this just referring to the minimization in 4 or something additional, if additional how is this different from 4?

The internal check does not refer to the minimization from eq.3, it refers to eq. 6 as discussed in this section after the introduction to the equation 6. Clarified in the text as:
To provide the most reliable flux estimate for the target observed by the BBR telescopes during operations, the three fluxes retrieved from the radiances co-registered at $H_{sw}$ undergo an internal consistency check to assess discrepancies between them. These fluxes are then merged, incorporating the uncertainties from the ADM construction and the observed deviations between the fluxes[...]When all $F_{sw}^i$ show discrepancies < 10%, $\delta^i = 1$ for all i. When two $F_{sw}^i$ agree to within ±10%, $\delta^i = 1$ with the outlier getting $\delta^i = 0$. If all $F_{sw}^i$ show fractional errors > 10%, only the lowest $\mathcal{E}_{Fsw}^i$ π $\mathcal{E}_{Lsw}^i$ uses $\delta^i = 1$, and $\delta^i = 0$ for the others.

Line 222 'higher cloud those observed'... not sure if you mean 'higher cloud than those observed' or something else.

Thanks, changed to: The MSI CTH is employed to detect either clouds in the surroundings of the nadir clear-sky domain or higher clouds than those observed in the nadir domain.

Line 235 add to last case 'and $\delta = 0$ for the others'.

Thanks, added at the end of the sentence: If all $F_{sw}^i$ show fractional errors > 10%, only the lowest $\mathcal{E}_{Fsw}^i$ π $\mathcal{E}_{Lsw}^i$ uses $\delta^i = 1$, and $\delta^i = 0$ for the others.

Line 242 Add reference for the GERB issue with semi-transparent cloud issues: Dewitte et al 2008 https://doi.org/10.1016/j.asr.2007.07.042 and Clerbaux et al 2009 https://doi.org/10.1016/j.rse.2008.08.016.

Thanks for the suggestion, added both references.

Line 247 to 250. I think it worth noting here again the absence of a water vapour channel and that the channel difference fills that role.

Added the following sentence: Although there is no water vapour channel in the MSI, the use of the thermal channels difference overcomes this lack.

Line 252 'allows to obtain' → 'enable sufficiently accurate anisotropy models to be derived from...'

Thanks for the suggestion. Modified accordingly: The lower anisotropy in the LW domain, with lower errors expected in the inversion process (Bodas, 2002), enables sufficiently accurate anisotropy models to be derived from radiative transfer (RT) simulations.

Line 252  Do we have any idea what the theoretical error is or any reference of what is possible from theory any reference for this assertion even if it is the LW errors achieved in GERB c.f. CERES for clear sky?

The theoretical error estimated in Bodas et al 2003 is lower than 3 Wm$^{-2}$ for clear sky scenes and lower than 6 Wm$^{-2}$ for overcast scenes for nadir. For VZA ~50° the errors are reduced for all cloud cover and they are lower than 1.5 Wm$^{-2}$. Added reference to Bodas thesis on the paper.

Line 253, might be worth stating that azimuthal dependence is negliglbe/neglected before stating R only is a function of VZA.

Thanks for your comment, the text has been modified as follows: "Although some studies (Minnis and Khaiyer, 2000 and Minnis 2004) have shown that under certain situations longwave radiances can strongly depend on the azimuth angle, our simulations have assumed azimuthal symmetry of the longwave radiation."

Line 256 can you put Velazquez et al 2010 on zelando and get a doi to make it accessible?

Link to the dataset and description is now available in the references.

Line 270  'done at'→'obtained at a' / Line 271  'would'→'should' or 'will'  / Line 274  'and statistically reducing the'→'reducing the'

Thanks, corrected as suggested.

Line 281 'in average' →'on average'

Thanks, changed to: This means that in the real world the thermal flux is on average more dependent on 3-dimensional effects than on plane parallel ones.

Line 286 Can you state what cloud properties are used, just fraction and height or something more?

Clarified in the text: "MSI radiances, brightness temperatures, and cloud retrievals (i.e., cloud fraction and CTH), which serve as input variables for flux retrieval models, are averaged over the different BBR resolutions (standard, small, full and assessment)."

Line 303 'chained…to'→ 'interfaces…with'

Thanks, changed to: "The BMA-FLX processor interfaces, within EarthCARE processing scheme, with M-CLD to receive the MSI-based CTH and cloud mask from M-COP and M-CM, to M-RGR to receive MSI regridded radiances and brightness temperatures, to A-LAY to receive the ATLID-based cloud top height from A-CTH, to BM-RAD to receive unfiltered radiances, to X-MET to receive high-resolution forecasts, and to ACM-RT (Cole et al., 2023) to provide co-registered radiances and fluxes."

Line 349 'As expected' is it expected in the model case specifically because it is using the plane parallel assumption can you specify that.

It is expected that error in the radiance to flux inversion is smaller for the fore and aft views of the BBR given the lower anisotropy of the radiance field for the off-nadir views. Text clarified and reference added: "As expected, given the lower anisotropy of the radiance field for the off-nadir views (Suttles et al., 1989), the aft and fore views present the lowest differences with respect to the model truth in the LW fluxes."

Line 354 Can you clarify here and in the caption to figure 5 and figure 6 if the BMA-FLX results shown are the combined view result.

Thanks, but we think that this was already mentioned in the text and in the captions.

Line 358 'employed both' → 'both employed'

Thanks, as X-MET has also been added it has been changed to all: "The cloud cover from the model truth (ranging from clear to overcast), the surface classification from GLCC and new X-MET, all employed in the ADM scene identification (e.g., water bodies, forests, savannahs, etc.), are presented at the bottom of the plots."

Line 381 'the longwave results remain consistent...' → 'the longwave results are identical to' (?)

Thanks, changed to: "The LW results remain consistent with those in the previous section because the LW algorithm utilizes broad and narrow-band radiances independently of L2 cloud retrievals."

Line 420 'greater dependency' → 'dependency' (given the lw has no dependency).

Thanks, changed to: "In contrast, the SW fluxes showed greater deviations from the model truth, primarily due to the more complex anisotropy of solar radiation footprints and their dependence on cloud-retrieved fields."

Line 486 – The doi associated with this paper is for the unpublished version I think it needs to be updated to https://doi.org/10.5194/amt-16-5327-2023 unless you specifically want to reference the unpublished submitted version and not the final publication.

Thanks, updated to: Donovan, D. P., Kollias, P., Velázquez Blázquez, A., and van Zadelhoff, G.-J.: The Generation of EarthCARE L1 Test Data sets Using Atmospheric Model Data Sets, EGUsphere, 2023, 1–54, https://doi.org/10.5194/amt-16-5327-2023.

---

## Referee Report (RR1)

I thank the authors for their efforts to respond to the original review and clarify my questions which was done appropriately. What I list below are changes of a technical nature mostly in respect of the additional text and changes. Line numbers refer to the revised version line numbers.

Line 72 Changes to this section are fine in general and address the original confusion but for complete correctness I suggest amending 'The thermal flux has the same dependencies as the solar flux except for $\theta_0$,' to 'The thermal flux has the same form but without the dependence on the solar enith ($\theta_0$) and with $\phi$ denoting the view azimuth rather than relative azimuth.'

The definition given in lines 114-117 for scene definition states angular geometry of the ADM is included in the scene definition and scene class is the part of the scene definition which determines the number of ADMs. This is fine but the rest of the discussion needs to be consistent with this definition but I think that scene definition is used several times in this section when you mean scene class. E.G. line 118 'scene definition' should be changed to 'scene classes' (with 'This classiiction' on line 121 changes to 'These classes' for consistency. Similarly table 1 should actually be titled Scene Classes rather than scene definition. And this should be reflected when referencing is (line 127).

Line 189 to 191.Change 'In clear-sky scenarios, the retrieval algorithm employs the two surface types with highest coverage within the BBR pixel, defining the observation as a mixed scene' to 'For clear sky cases which are a mix of surface types the observation is defined as mixed scene and the retrieval algorithm employs the two surface types with highest coverage within the BBR pixel'. Otherwise it sounds like you are defining all clear sky as mixed which presumably some might be a single surface type.

Lines 235 to 236. Given we have not yet discussed the LW this assertion, without any context or reference that the CTH is reliable for the thermal seems a bit odd. I would suggest removing 'The 90th percentile of the CTH derived from the MSI brightness temperature (BT), referred to as M-COP (Hünerbein et al., 2023a), is a reliable estimator for co-registering the BBR radiances in the thermal regime.' And just start the paragraph with the next sentence.

Lines 266 to 267. $\theta_{obl}$ needs to be defined. I think this needs to be rewritten to make the paragraph self-consistent, with some explanation as to why di starts at 1 and not for example zero. You are essentially saying that the minimum cloud height is 1/ tan ($\theta_{obl}$), (so 0.7km assuming 55 is the angle), however you just stated that the cloud could be anywhere from surface (which I assume is zero) to the tropopause so the assertion that d starts at 1 is at odds with this.  I think maybe you are saying that an image is only considered to be affected by parallax when there is some minimum difference between observed cloud and surface or nadir cloud height and oblique cloud height but this is not clear and in any case the latter would make d range from nadir CTH + something not 1. I think some correction is required here.

Lines 273 to 274. I think that selection of the excluded data reserved for testing as described in your reply to my question on this is fine. However I think calling it 'an entirely independent source' is incorrect and confusing making it sound like an entirely different dataset from CERES. I suggest replaincg 'Notably, this dataset for evaluation is an entirely independent source, as it was not used during the training of the ADM' with 'All the data in the evaluation dataset was excluded from the original training process'. Or 'The evaluation dataset represents a randomly selected subset of the originally identified potential training data that was excluded from the

training process'. I think it would be helpful to add one more sentence stating the RMS and bias errors found with this validation test.

Lines 292 to 294 Change: 'Even though in theory a multi-spectral model should be able to correctly handle all scene types, in practice it was demonstrated from previous GERB studies (Dewitte et al., 2008; Clerbaux et al., 2009) that a large bias was introduced in the case of semi-transparent clouds, fact that was indeed verified in the early stages of the selection and validation of the algorithm'. 'Previous studies for GERB (Dewitte et al., 2008; Clerbaux et al., 2009) has shown that using a single multi-spectral regression for all scenes can cause large biases for semi-transparent cloud. This problem was also highlighted in the early stages of the selection on and validation of the FMA-FLX processor algorithms'

Line 326 to 326 'To capitalize on this, the LW merging algorithm assigns a greater weight to the fore and aft views for plane-parallel scenes.' But you don't seem to identify plane parallel scenes and later you don't assign greater weight to fore and aft views so I think you mean "Thus plane parallel assumptions would indicate that greater weight should be placed on the fore and after views.'

Lines 322 to 323, "The three BBR thermal unfiltered radiances are co-registered at a reference level defined by the percentile 90th of the MSI CTH as described in section 2.1.5" Either specify for cloudy scenes or make general to all scenes by changing to "The three BBR thermal unfiltered radiances use the default surface co-registration for clear sky and for cloudy scenes are co-registered at a reference level defined by the percentile 90th of the MSI CTH as described in section 2.1.5" or "For cloudy scenes the three BBR thermal unfiltered radiances are co-registered at a reference level defined by the percentile 90th of the MSI CTH as described in section 2.1.5 instead of the default co-registration used for clear sky."

Line 362 'Proper validation...' to 'Full validation..'

Line 374 'allows for the identification...' to 'allows the identification..'

Lines 377 to 380. You state discrepancy 'between the radiative transfer simulated used in the simulated geophysical data and the modelled EarthCARE products' and later 'discrepancies in surface definitions between RTC model and the BMA_FLX processor', but don't address the possible (likely if plane parallel) discrepancy between the simulated EarthCARE flux 'truth' and real non real world fluxes until much later. I suggest adding a sentence on this here as it is a major limitation you have with what you need to work with and is very relevant for considering what your results mean.

Line 430to 431 "The combined approach shows the greatest advantage in terms of error metrics, with lower values compared to the view-based flux estimations" to "The combined approach results in better error metrics than the those for the induvial views'

Line 432 "...the combined approach has significant lower error metrics than the individual fluxes, indicating better overall agreement." To "...the combined approach results in significantly better error metrics than those of the individual fluxes, indicating better overall agreement."

Line 435 " indicating the superior overall performance" to ".indicating superior overall performance"

Line 449 to 450 ' This comparison is shown as a "pre-launch" numerical assessment experiment for validating the performance and reliability of the BMA-FLX processor in diverse environmental

conditions.' Add a sentence along the lines or "Within the limitations of the accuracy of the simulated fluxes provided as truth.'

Line 456 to 457 "Consequently, SW fluxes, which are obtained from ADMs constructed using satellite measurements, retrieved in the regions with simulated broken clouds tend to be noisy and less reliable." This sentence seems to imply that your empirical ADMs are wrong, noisy and unreliable, rather than just not representative of the unrealistic plane parallel situation of the simulated truth I thin you need to rephrase this to make sense for example "Consequently, fluxes from BMA-FLX, which are obtained from ADMs constructed using satellite measurements of the real world non-plane parallel anisotropy will not follow the radiance to flux relationships found in plane parallel simulations used as the truth in this study." This applies to both the SW and LW I don't see a need to restrict it to the SW.

Lines 457 to 458 "In the LW estimates, this results in noisy flux retrievals that are flattened out when increasing the averaging region." Whilst the LW estimate c.f. to the 'truth' do appear noisy and the differences do average out I don't know if there is evidence that this is plane parallel in origin or even a model/real world cause. I suggest just changing these sentence to something more vague such as "LW flux retrievals appear to have a high frequency variations not observed in the simulations that are flattened out with increasing avering region". The issue with your longwave fluxes occurs in some clear sky cases and some cloud. It just appears that your fluxes are considerably more sensitive to some input that is varying on these high frequency scales that the simulated fluxes are not seeing. Either due to changing between regressions or I suppose the use of your $z_2$ as a channel difference but you would need to look at your inputs to see where that is coming from. In any case I suggest just changing the sentence to not imply a known cause as suggested.

Line 477 'significantly impact in' to 'significantly impact'

Line 478 "The combined results for Baja do not differ significantly from the previous analysis." This statement does not match with what is shown in the tables or the plots, they seem very significantly different to me. I think you are not considering changes made to the values shown in table 3 presented here from those shown in the original submission. Please correct this statement or the results as appropriate.

Line 479 to 480 "This is primarily due to …" "The average values are also influenced by ." I think it is far from clear is this is primary cause but it clearly influences comparison of the average values.

Line 481 to 482 "Overall, the algorithm for combining the view-based fluxes performs exceptionally well in mitigating the impact of incorrect retrievals from the nadir, aft, and fore models." If the 'incorrect retrievals' here are meant to mean the cloud retrievals you are discussing then this isn't very convincing for both Halifax and Baja the improvement for combination here seems very similar to what was obtained when you combined them without cloud retrieval errors. You do get a very significant reduction for Hawaii but given the similar strange inconsistency between views you presented for Baja in the original manuscript that seems to have gone away in this revision I wonder if maybe you should double check these Hawaii results and see if they have a a similar problem. I think this sentence needs to be rephrased to talk generally about the combination improvement rather than specifically for the effect of cloud retrieval errors although obviously helpful for those too.

518 to 520: "Instances where the error metrics exceeded the 10 Wm−2 threshold suggest that achieving the radiative closure goal might be challenging, highlighting the complexity of meeting the mission's objectives and underscoring the ambitious accuracy requirements" I would rephrase this and possibly move your discussion in lines 527 to 530 about the simulated environment here to add context. For example change to "Instances where the error metrics exceeded the 10 Wm−2 threshold may indicate that achieving the radiative goal will be challenging requiring both improvements to the cloud property retrieval and the BMA-FLX algorithm. However, this needs to be considered in the context of the simulated environment used for this study and the likely inaccuracy of the simulated fluxes used for 'truth' here particularly for broken cloud conditions. Further validation of the BMA-FLX will occur during the commissioning ……"

Line 638 – DOI is for preprint needs be updated to published version.

---

## Author Response (AR2)

The authors would like to thank Dr. Jacqueline Russell for her comments and corrections. Please find below our replies in blue.

**Responses to Referee #2 Jacqueline Russell**

I thank the authors for their efforts to respond to the original review and clarify my questions which was done appropriately. What I list below are changes of a technical nature mostly in respect of the additional text and changes. Line numbers refer to the revised version line numbers.

Line 72 Changes to this section are fine in general and address the original confusion but for complete correctness I suggest amending 'The thermal flux has the same dependencies as the solar flux except for $\theta_0$,'to 'The thermal flux has the same form but without the dependence on the solar zenith ($\theta_0$) and with $\phi$ denoting the view azimuth rather than relative azimuth.'

Agreed, changed as suggested.

The definition given in lines 114-117 for scene definition states angular geometry of the ADM is included in the scene definition and scene class is the part of the scene definition which determines the number of ADMs. This is fine but the rest of the discussion needs to be consistent with this definition but I think that scene definition is used several times in this section when you mean scene class. E.G. line 118 'scene definition' should be changed to 'scene classes' (with 'This classification' on line 121 changes to 'These classes' for consistency. Similarly table 1 should actually be titled Scene Classes rather than scene definition. And this should be reflected when referencing is (line 127).

Thank you for pointing out the inconsistency. The text has been revised accordingly.

Line 189 to 191.Change 'In clear-sky scenarios, the retrieval algorithm employs the two surface types with highest coverage within the BBR pixel, defining the observation as a mixed scene' to 'For clear sky cases which are a mix of surface types the observation is defined as mixed scene and the retrieval algorithm employs the two surface types with highest coverage within the BBR pixel'. Otherwise it sounds like you are defining all clear sky as mixed which presumably some might be a single surface type.

Agreed, changed to "For clear-sky cases, which can be a mix of different surface types, the observation is defined as mixed scene and the retrieval algorithm employs the two surface types with highest coverage within the BBR pixel."

Lines 235 to 236. Given we have not yet discussed the LW this assertion, without any context or reference that the CTH is reliable for the thermal seems a bit odd. I would suggest removing 'The 90th percentile of the CTH derived from the MSI brightness temperature (BT), referred to as MCOP (Hünerbein et al., 2023a), is a reliable estimator for co-registering the BBR radiances in the thermal regime.' And just start the paragraph with the next sentence.

Agreed, sentence removed.

Lines 266 to 267. $\theta_{obl}$ needs to be defined. I think this needs to be rewritten to make the paragraph self-consistent, with some explanation as to why di starts at 1 and not for example zero. You are essentially saying that the minimum cloud height is 1/ tan ($\theta_{obl}$), (so 0.7km assuming 55 is the angle), however you just stated that the cloud could be anywhere from surface (which I assume is zero) to the tropopause so the assertion that d starts at 1 is at odds with this. I think maybe you are saying that an image is only considered to be affected by parallax when there is some

minimum difference between observed cloud and surface or nadir cloud height and oblique cloud height but this is not clear and in any case the latter would make d range from nadir CTH + something not 1. I think some correction is required here.

Agreed, $\theta_{obl}$ has been defined.

As for $d_i$, the along-track sampling is 1km. The default co-registration is at surface, next and previous samples are 1km away.

Lines 273 to 274. I think that selection of the excluded data reserved for testing as described in your reply to my question on this is fine. However I think calling it 'an entirely independent source' is incorrect and confusing making it sound like an entirely different dataset from CERES. I suggest replaincg 'Notably, this dataset for evaluation is an entirely independent source, as it was not used during the training of the ADM' with 'All the data in the evaluation dataset was excluded from the original training process'. Or 'The evaluation dataset represents a randomly selected subset of the originally identified potential training data that was excluded from the training process'. I think it would be helpful to add one more sentence stating the RMS and bias errors found with this validation test.

Thanks for your contribution. This has been clarified in the text accordingly.

Lines 292 to 294 Change: 'Even though in theory a multi-spectral model should be able to correctly handle all scene types, in practice it was demonstrated from previous GERB studies (Dewitte et al., 2008; Clerbaux et al., 2009) that a large bias was introduced in the case of semitransparent clouds, fact that was indeed verified in the early stages of the selection and validation of the algorithm'. 'Previous studies for GERB (Dewitte et al., 2008; Clerbaux et al., 2009) has shown that using a single multi-spectral regression for all scenes can cause large biases for semi-transparent cloud. This problem was also highlighted in the early stages of the selection on and validation of the FMA-FLX processor algorithms'

Agreed, proposed change has been corrected in the text.

Line 326 to 326 'To capitalize on this, the LW merging algorithm assigns a greater weight to the fore and aft views for plane-parallel scenes.' But you don't seem to identify plane parallel scenes and later you don't assign greater weight to fore and aft views so I think you mean "Thus plane parallel assumptions would indicate that greater weight should be placed on the fore and after views.'

Thanks for the suggestion. Modified in the text.

Lines 322 to 323, "The three BBR thermal unfiltered radiances are co-registered at a reference level defined by the percentile 90th of the MSI CTH as described in section 2.1.5" Either specify for cloudy scenes or make general to all scenes by changing to "The three BBR thermal unfiltered radiances use the default surface co-registration for clear sky and for cloudy scenes are co-registered at a reference level defined by the percentile 90th of the MSI CTH as described in section 2.1.5" or "For cloudy scenes the three BBR thermal unfiltered radiances are coregistered at a reference level defined by the percentile 90th of the MSI CTH as described in section 2.1.5 instead of the default co-registration used for clear sky."

Agreed, changed to "The three BBR thermal unfiltered radiances use the default surface co-registration for clear sky scenes but in presence of clouds are co-registered at a reference level defined by the percentile 90th of the MSI CTH from the M-COP product (Hünerbein et al., 2024)."

Line 362 'Proper validation…' to 'Full validation..'

Corrected.

Line 374 'allows for the identification…'to 'allows the identification..'

Corrected

Lines 377 to 380. You state discrepancy 'between the radiative transfer simulated used in the simulated geophysical data and the modelled EarthCARE products' and later 'discrepancies in surface definitions between RTC model and the BMA_FLX processor', but don't address the possible (likely if plane parallel) discrepancy between the simulated EarthCARE flux 'truth' and real non real world fluxes until much later. I suggest adding a sentence on this here as it is a major limitation you have with what you need to work with and is very relevant for considering what your results mean.

Agreed, sentence added in the document.

Line 430 to 431 "The combined approach shows the greatest advantage in terms of error metrics, with lower values compared to the view-based flux estimations" to "The combined approach results in better error metrics than the those for the induvial views'

Corrected.

Line 432 "…the combined approach has significant lower error metrics than the individual fluxes, indicating better overall agreement." To "…the combined approach results in significantly better error metrics than those of the individual fluxes, indicating better overall agreement."

Corrected.

Line 435 " indicating the superior overall performance" to ".indicating superior overall performance"

Corrected.

Line 449 to 450 ' This comparison is shown as a "pre-launch" numerical assessment experiment for validating the performance and reliability of the BMA-FLX processor in diverse environmental conditions.' Add a sentence along the lines or "Within the limitations of the accuracy of the simulated fluxes provided as truth.'

Thanks for the suggestion, however, we believe that adding this extra information to the sentence does not help the reader to better understand the context and the accuracy of the simulated fluxes are difficult to quantify.

Line 456 to 457 "Consequently, SW fluxes, which are obtained from ADMs constructed using satellite measurements, retrieved in the regions with simulated broken clouds tend to be noisy and less reliable." This sentence seems to imply that your empirical ADMs are wrong, noisy and unreliable, rather than just not representative of the unrealistic plane parallel situation of the simulated truth I think you need to rephrase this to make sense for example "Consequently, fluxes from BMA-FLX, which are obtained from ADMs constructed using satellite measurements of the real world non-plane parallel anisotropy will not follow the radiance to flux relationships found in plane parallel simulations used as the truth in this study." This applies to both the SW and LW I don't see a need to restrict it to the SW.

Thanks for your comment. This justification has been included in the text. But please note that this does not apply to LW as the thermal ADMs are based on results from plane-parallel radiative transfer simulations.

Lines 457 to 458 "In the LW estimates, this results in noisy flux retrievals that are flattened out when increasing the averaging region." Whilst the LW estimate c.f. to the 'truth' do appear noisy and the differences do average out I don't know if there is evidence that this is plane parallel in origin or even a model/real world cause. I suggest just changing these sentence to something more vague such as "LW flux retrievals appear to have a high frequency variations not observed in the simulations that are flattened out with increasing avering region". The issue with your longwave fluxes occurs in some clear sky cases and some cloud. It just appears that your fluxes are considerably more sensitive to some input that is varying on these high frequency scales that the simulated fluxes are not seeing. Either due to changing between regressions or I suppose the use of your z2 as a channel difference but you would need to look at your inputs to see where that is coming from. In any case I suggest just changing the sentence to not imply a known cause as suggested.

Agreed and rewritten as suggested.

Line 477 'significantly impact in' to 'significantly impact'

Thanks, corrected.

Line 478 "The combined results for Baja do not differ significantly from the previous analysis." This statement does not match with what is shown in the tables or the plots, they seem very significantly different to me. I think you are not considering changes made to the values shown in table 3 presented here from those shown in the original submission. Please correct this statement or the results as appropriate.

Agreed, rewritten in the sentence.

Halifax and Hawaii conclusions are still valid, but not those for Baja, which have been properly addressed.

Line 479 to 480 "This is primarily due to …""The average values are also influenced by ." I think it is far from clear is this is primary cause but it clearly influences comparison of the average values.

Agreed, corrected in text.

Line 481 to 482 "Overall, the algorithm for combining the view-based fluxes performs exceptionally well in mitigating the impact of incorrect retrievals from the nadir, aft, and fore models." If the 'incorrect retrievals' here are meant to mean the cloud retrievals you are discussing then this isn't very convincing for both Halifax and Baja the improvement for combination here seems very similar to what was obtained when you combined them without cloud retrieval errors. You do get a very significant reduction for Hawaii but given the similar strange inconsistency between views you presented for Baja in the original manuscript that seems to have gone away in this revision I wonder if maybe you should double check these Hawaii results and see if they have a a similar problem. I think this sentence needs to be rephrased to talk generally about the combination improvement rather than specifically for the effect of cloud retrieval errors although obviously helpful for those too.

Thanks for your comment, the Hawaii main problem was, as stated in the paper, related to the cloud retrieval algorithm. However, Baja and Halifax problem is mainly related to the non-coherent classification of the surface and cloud properties over snow. We believe no change is needed here as indeed the combining algorithm performs well and mitigates the use of incorrect cloud retrievals and/or surface properties.

518 to 520: "Instances where the error metrics exceeded the 10 Wm−2 threshold suggest that achieving the radiative closure goal might be challenging, highlighting the complexity of meeting the mission's objectives and underscoring the ambitious accuracy requirements" I would rephrase this and possibly move your discussion in lines 527 to 530 about the simulated environment here to add context. For example change to "Instances where the error metrics exceeded the 10 Wm−2 threshold may indicate that achieving the radiative goal will be challenging requiring both improvements to the cloud property retrieval and the BMA-FLX algorithm. However, this needs to be considered in the context of the simulated environment used for this study and the likely inaccuracy of the simulated fluxes used for 'truth' here particularly for broken cloud conditions. Further validation of the BMA-FLX will occur during the commissioning ……"

Agreed, rewritten.

Line 638 – DOI is for preprint needs be updated to published version.

We understand it refers to line 633. Thanks, fixed.